# Nanoscaled RIM clustering at presynaptic active zones revealed by endogenous tagging

Achmed Mrestani[1,2,3,*] , Sven Dannhäuser[1,*] , Martin Pauli[1], Philip Kollmannsberger[4], Martha Hübsch[1] , Lydia Morris[3], Tobias Langenhan[3] , Manfred Heckmann[1] , Mila M Paul[1,5]

**Chemical synaptic transmission involves neurotransmitter release from presynaptic active zones (AZs). The AZ protein Rab-3-interacting molecule (RIM) is important for normal $Ca^{2+}$-triggered release. However, its precise localization within AZs of the glutamatergic neuromuscular junctions of *Drosophila melanogaster* remains elusive. We used CRISPR/Cas9-assisted genome engineering of the *rim* locus to incorporate small epitope tags for targeted super-resolution imaging. A V5-tag, derived from simian virus 5, and an HA-tag, derived from human influenza virus, were N-terminally fused to the RIM Zinc finger. Whereas both variants are expressed in co-localization with the core AZ scaffold Bruchpilot, electrophysiological characterization reveals that AP-evoked synaptic release is disturbed in $rim^{V5-Znf}$ but not in $rim^{HA-Znf}$. In addition, $rim^{HA-Znf}$ synapses show intact presynaptic homeostatic potentiation. Combining super-resolution localization microscopy and hierarchical clustering, we detect ~10 $RIM^{HA-Znf}$ subclusters with ~13 nm diameter per AZ that are compacted and increased in numbers in presynaptic homeostatic potentiation.**

## Introduction

Neurotransmitter release at presynaptic active zones (AZs) is fundamental for information processing within the nervous system (Südhof, 2012; Neher & Brose, 2018; Emperador-Melero & Kaeser, 2020). The AZ scaffold comprises a core set of proteins precisely arranged within nanometers. Modifications of this nanoarchitecture contribute to functional changes and the diversification of synaptic strength (Atwood & Karunanithi, 2002; Mrestani et al, 2021). Among these proteins, the multi-domain protein Rab3-interacting molecule (RIM) is essential for docking of synaptic vesicles in mammals (Fernández-Busnadiego et al, 2013) and for the clustering of voltage-gated calcium channels (VGCCs) close to release sites (Han et al, 2011;

Kaeser et al, 2011). The multi-domain structure of RIM enables multiple interactions: the N-terminal zinc finger binds to Rab3 and Munc13-1 (Wang et al, 1997; Andrews-Zwilling et al, 2006; Deng et al, 2011), the PxxP motif to RIM-binding protein (RIM-BP, Wang et al, 2000; Hibino et al, 2002) and the PDZ-, $C_2A$, and $C_2B$ domains to VGCCs (Coppola et al, 2001; Kiyonaka et al, 2007; Miki et al, 2007; Kaeser et al, 2011, 2012).

At the *Drosophila melanogaster* neuromuscular junction (NMJ), RIM promotes VGCC accumulation within the AZ, controls the readily releasable pool of synaptic vesicles, and is essential for presynaptic homeostatic plasticity (Graf et al, 2012; Müller et al, 2012; Paul et al, 2022), and thus the dynamic regulation of synaptic strength (Davis & Müller, 2015). Whereas remarkable reorganization of RIM during synaptic plasticity was described in cultured murine neurons (Tang et al, 2016; Müller et al, 2022), the nanoscale organization of RIM at *Drosophila* AZs, remains unclear. Previous confocal and stimulation emission depletion imaging relied on overexpression or the endogenous expression of GFP-fused constructs (Graf et al, 2012; Petzoldt et al, 2020). During the last decade, various tools for the generation of genetically marked constructs emerged, offering new labeling strategies for super-resolution imaging.

Here, we used the clustered regularly interspaced short palindromic repeats (CRISPR)/CRISPR-associated protein 9 (Cas9) system (Bassett et al, 2013; Gratz et al, 2013, 2014; Kondo & Ueda, 2013; Yu et al, 2013; Port et al, 2014) to introduce small epitope tags into RIM to enable its precise localization within presynaptic AZs. We inserted either a V5-tag derived from the P and V proteins of the simian virus 5 (GKPIPNPLLGLDST, Hanke et al, 1992) or an HA-tag derived from the human influenza virus (YPYDVPDYA) into the endogenous *rim* ORF to tag the N-terminal end of the zinc finger domain ($rim^{V5-Znf}$ and $rim^{HA-Znf}$). Both tags are small and low in molecular weight; however, they substantially differ in their iso-electric points and the number of charged residues. As the zinc finger domain of RIM entertains crucial molecular contacts with Rab3 and therefore vesicle binding, it was essential to test if the epitope tags interfere with the physiological function of RIM and, by

[1]Department of Neurophysiology, Institute of Physiology, University of Würzburg, Würzburg, Germany   [2]Department of Neurology, Leipzig University Medical Center, Leipzig, Germany   [3]Division of General Biochemistry, Rudolf Schönheimer Institute of Biochemistry, Medical Faculty, Leipzig University, Leipzig, Germany   [4]Biomedical Physics, Heinrich Heine University Düsseldorf, Düsseldorf, Germany   [5]Department of Orthopedic Trauma, Hand, Plastic and Reconstructive Surgery, University Hospital of Würzburg, Würzburg, Germany

Correspondence: heckmann@uni-wuerzburg.de; mila.paul@uni-wuerzburg.de
*Achmed Mrestani and Sven Dannhäuser contributed equally to this work

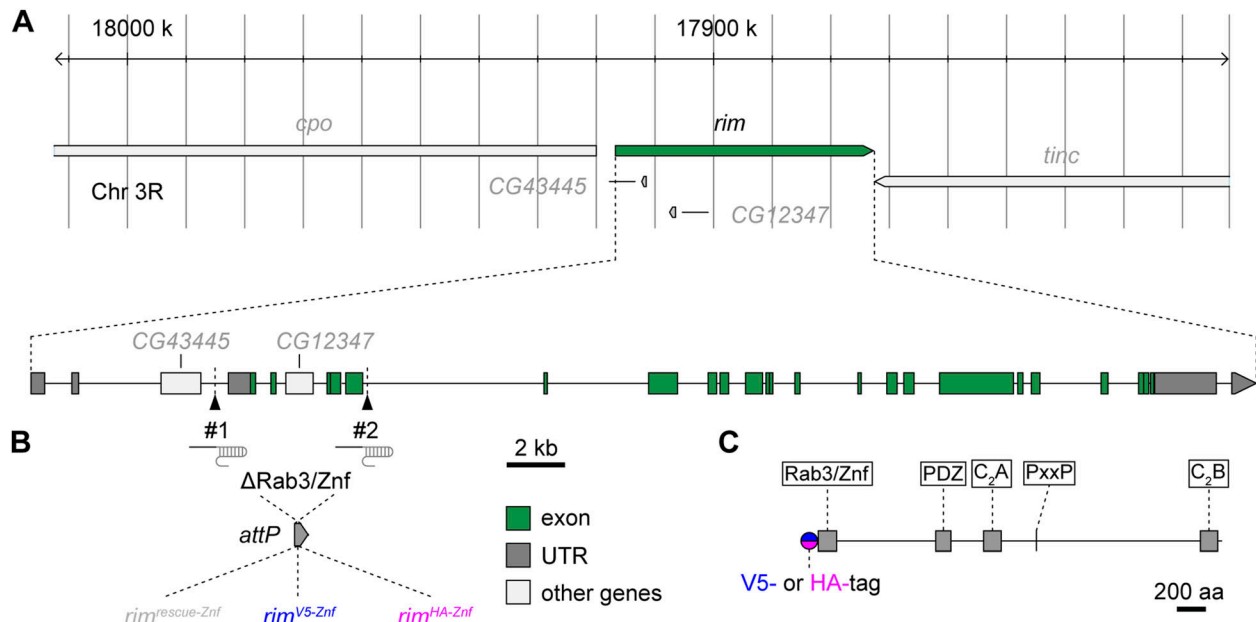

**Figure 1. Construction of epitope-tagged *rim* alleles using CRISPR/Cas9.**
**(A)** Overview of the *Drosophila melanogaster rim* locus on chromosome III and illustration of genomic targeting strategy. #1 and #2 mark positions of the *gRNA* probes. **(B)** Location and size of the *rim* gene fragment excised by CRISPR/Cas9 and replaced by φC31-mediated integration to generate *rim*[rescue-Znf] (gray), *rim*[V5-Znf] (blue), and *rim*[HA-Znf] (magenta) alleles. UTR, untranslated region. **(C)** *Drosophila* RIM protein domain positions: Rab3/zinc finger (Znf), PDZ, $C_2A$, PxxP, and $C_2B$ domains and position of the N-terminal epitope tag (V5-Znf or HA-Znf) is indicated.

extension, neurotransmitter release. Electrophysiological characterizations revealed that the HA-tag, in contrast to the V5-tag, leaves spontaneous and AP-evoked synaptic release and the expression of presynaptic plasticity undisturbed. We found efficient expression of RIM[V5-Znf] and RIM[HA-Znf] in distinct co-localization with the AZ scaffold protein Bruchpilot (Brp) in the peripheral and central nervous systems of *Drosophila* larvae. Thus, we investigated the nanotopology of RIM[HA-Znf] at the AZ using two-color *direct* stochastic optical reconstruction microscopy (*d*STORM, Heilemann et al, 2008; van de Linde et al, 2011) in combination with hierarchical density-based spatial clustering of applications with noise (HDBSCAN). Our super-resolution approach reveals information about the RIM[HA-Znf] nano-assembly at the *Drosophila* AZ into distinct clusters of ~130 nm² size in ~120 nm distance from the AZ center. Furthermore, these RIM[HA-Znf] clusters change in response to an acute homeostatic challenge by shrinking, increasing in localization density, and in absolute numbers.

## Results

### CRISPR/Cas9-engineering of *rim*

Previous RIM imaging at the *Drosophila* NMJ utilized overexpression or endogenous expression of GFP-fused proteins (Graf et al, 2012; Petzoldt et al, 2020). In the latter, a recombinase-mediated cassette exchange-derived line based on the *Minos*-mediated integration cassette collection (Nagarkar-Jaiswal et al, 2015) targeting only one of the 14 predicted splice variants was employed (Petzoldt et al, 2020). Here, we genomically engineered alleles of the single *Drosophila rim* gene to investigate the population of RIM at AZ scaffolds at

levels under endogenous *cis*- and *trans*-regulatory control (Fig 1A). We combined CRISPR/Cas9-assisted genome editing (Gratz et al, 2014) and φC31-mediated recombinase-mediated cassette exchange (Huang et al, 2009) as previously applied to expedite generation of alleles encoding RIM $C_2A$ domain mutations (Paul et al, 2022). To this end, we targeted a genome fragment spanning exons 3–7 that cover the coding sequence for the Rab3-interacting/zinc finger (Znf) domain of RIM and replaced it by a rescue genomic fragment with the full removed sequence (*rim*[rescue-Znf]), or genomic fragments with additional coding sequences for the V5- or HA-tag before the common start codon (*rim*[V5-Znf] and *rim*[HA-Znf], respectively, Fig 1B), leading to N-terminally tagged fusion proteins (Fig 1C).

### Baseline synaptic transmission at rim[V5−Znf] and rim[HA−Znf] terminals

Next, we performed two-electrode voltage clamp recordings for evaluation of spontaneous and evoked synaptic transmission at larval NMJs of the epitope-tagged *rim* variants (Fig 2 and Tables S1 and S2). We recorded miniature excitatory postsynaptic currents (mEPSCs) to examine spontaneous release in wt and rim[rescue−Znf] third instar larvae to control for normal synaptic release in the latter. In addition, mEPSCs of rim[V5−Znf] and rim[HA−Znf] larvae were compared with rim[rescue−Znf] (Fig 2A, all homozygous). mEPSC amplitudes and the frequency of spontaneous fusion events were unchanged at both rim[V5−Znf] and rim[HA−Znf] NMJs (Fig 2B). Next, we measured evoked excitatory postsynaptic currents (eEPSCs) in response to nerve stimulation (Fig 2C). eEPSC amplitudes were unaltered in rim[rescue−Znf] compared with wt (Fig 2D). Interestingly, eEPSC amplitudes were significantly decreased in *rim*[V5−Znf] compared with

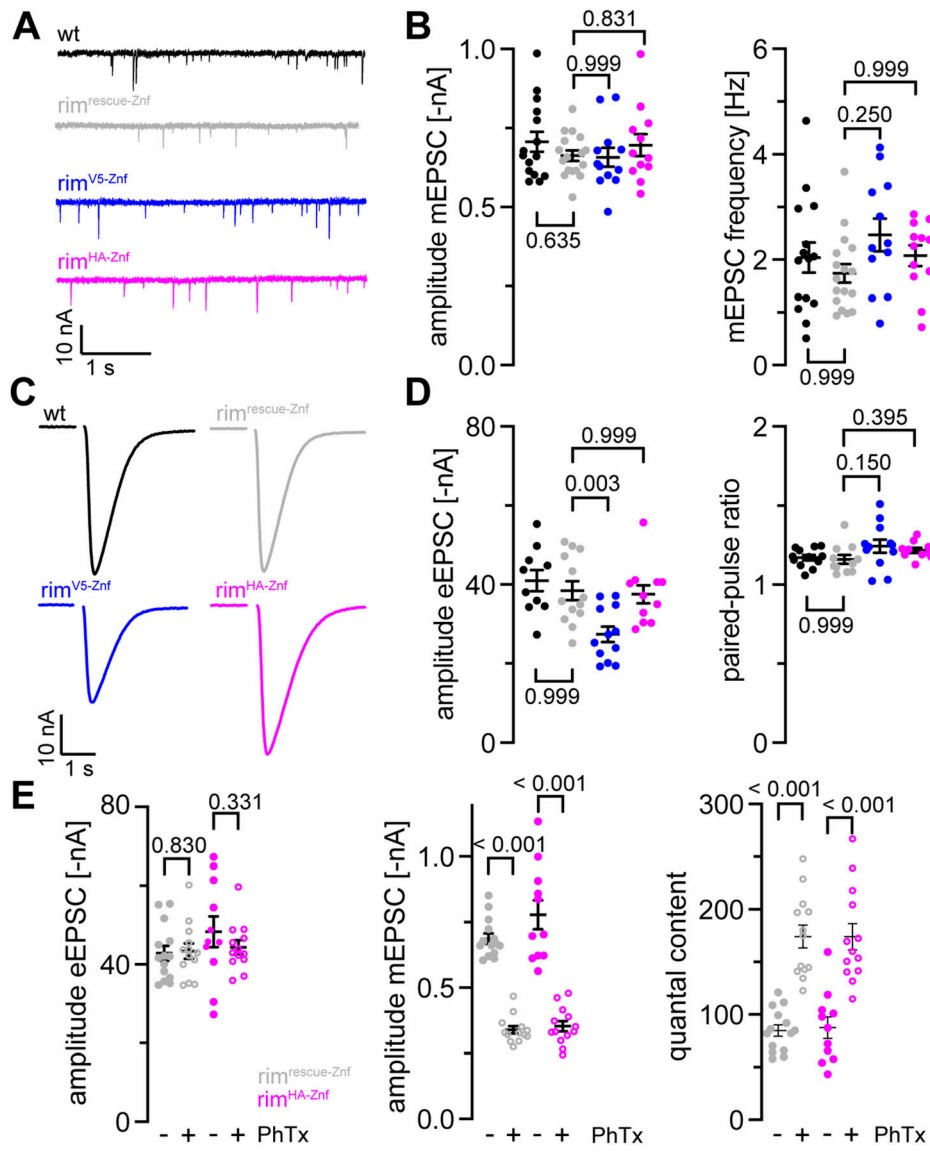

**Figure 2. Intact synaptic transmission at rim^HA−Znf neuromuscular junctions.**
**(A)** Miniature excitatory postsynaptic currents (mEPSCs) at WT (black), rim^rescue−Znf (gray), rim^V5−Znf (blue), and rim^HA−Znf (magenta) neuromuscular junctions (NMJs). **(B)** Mean ± SEM mEPSC amplitude and frequency in wt (n = 15 NMJs in seven animals), rim^rescue−Znf (n = 17/8), rim^V5−Znf (n = 12/6), and rim^HA−Znf (n = 12/5). Scatter plots show individual data points, individual *P*-values are indicated. **(C)** Evoked EPSCs (eEPSCs) samples at NMJs of the same four genotypes. **(D)** eEPSC amplitude and paired-pulse ratios with 30 ms interstimulus interval in wt (n = 10 NMJs in seven animals), rim^rescue−Znf (n = 13/7), rim^V5−Znf (n = 12/5), and rim^HA−Znf (n = 11/5). **(E)** Mean ± SEM eEPSC amplitude, mEPSC amplitude and quantal content in rim^rescue−Znf and rim^HA−Znf animals treated with PhTx in DMSO (+, open circles) or DMSO (−, filled circles). rim^HA−Znf larvae show undisturbed presynaptic homeostatic potentiation in response to PhTx stimulation (rim^rescue−Znf: 14 NMJs in seven animals in DMSO, 13/6 in PhTx; rim^HA−Znf: 11/5 in DMSO, 13/5 in PhTx).

rim^rescue−Znf, whereas they were unchanged in rim^HA−Znf (Fig 2D), indicating that the V5-tag but not the HA-tag interferes with evoked synaptic release. We also tested whether insertion of the tags alters synaptic short-term plasticity but found paired pulse ratios unchanged at both rim^V5−Znf and rim^HA−Znf NMJs (Fig 2D). Taken together, these data show that both spontaneous and evoked synaptic transmissions are intact in rim^rescue−Znf and rim^HA−Znf variants. Furthermore, as evoked synaptic release is significantly decreased in rim^V5−Znf but not in rim^HA−Znf, the HA-tag is an appropriate tool for further assessment of structure–function relationships at the *Drosophila* NMJ.

### Undisturbed presynaptic homeostatic potentiation (PHP) at rim^HA−Znf NMJs

RIM is required for PHP by modulation of the readily releasable vesicle pool (RRP, Müller et al, 2012). Thus, we probed if our genetically engineered *rim* variants carrying an epitope tag at the

N-terminal zinc finger domain still exhibit PHP at normal levels. To test if the HA-tagged RIM is still functional, we measured the electrophysiological response to an acute homeostatic challenge using Philanthotoxin (PhTx) in rim^HA−Znf animals compared with rim^rescue−Znf larvae (Fig 2E and Tables S3 and S4). We found that upon PhTx treatment, rim^HA−Znf showed the same increase in quantal content and thus evoked EPSC restoration as rim^rescue−Znf. This indicates that rim^HA−Znf NMJs exhibit unperturbed PHP. We conclude that genomic HA-tag insertion into the endogenous *rim* ORF leaves presynaptic plasticity intact at larval *Drosophila* NMJs.

### Expression of tagged *rim* alleles in the *Drosophila* nervous system

After verification that rim^HA−Znf but not rim^V5−Znf larvae show normal synaptic release upon AP-evoked stimulation and at homeostatic challenge, we investigated the expression of RIM with the N-terminally fused epitope tags within the *Drosophila* nervous system.

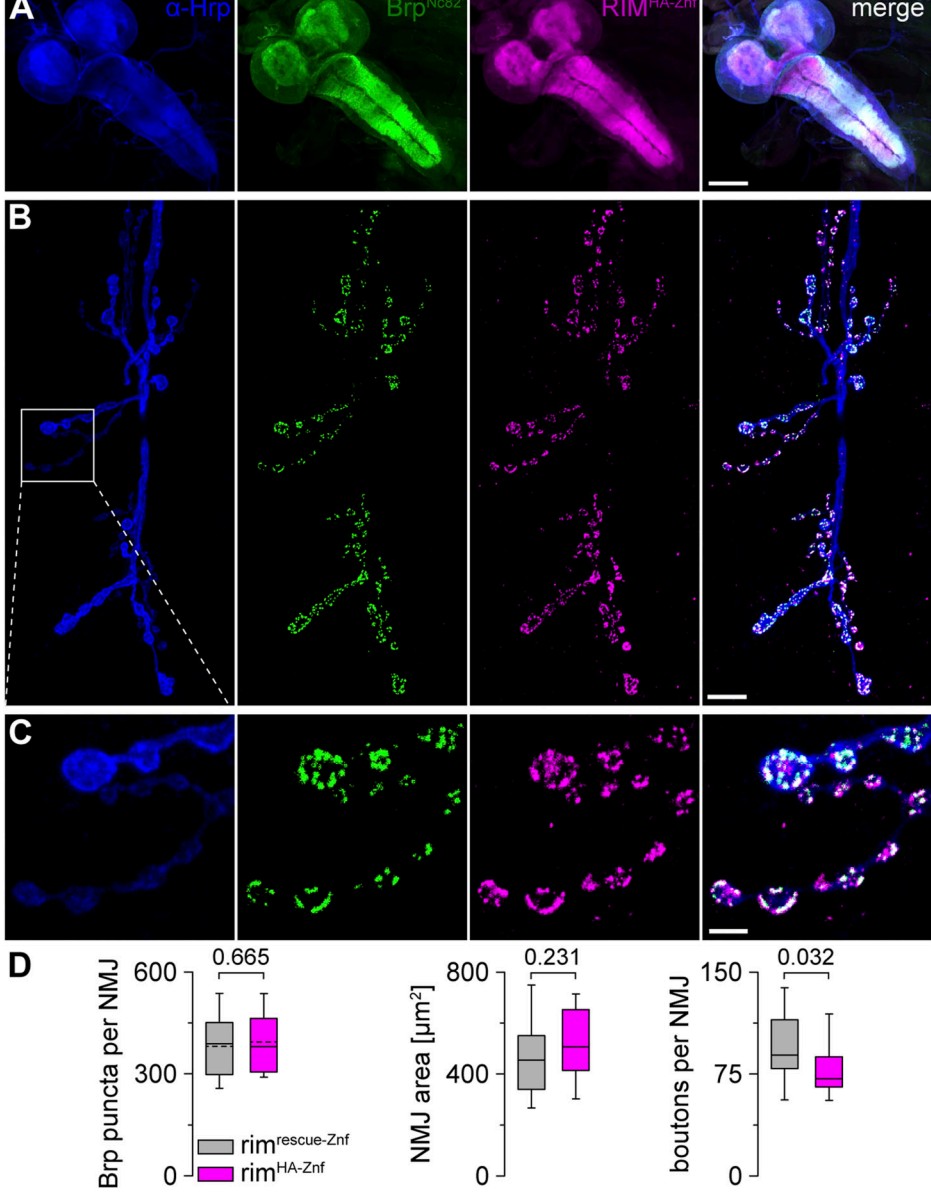

**Figure 3. Expression of RIM[HA−Znf] in the Drosophila central and peripheral nervous systems.**
**(A)** Confocal image of the ventral nerve cord of a male third instar larva stained with α-HRP (blue), Brp[Nc82] (green), and rabbit monoclonal α-HA antibody to visualize RIM[HA−Znf] (magenta). **(B, C)** α-HRP, Brp[Nc82], and RIM[HA−Znf] at a neuromuscular junction (NMJ) on abdominal muscles 6/7. White box highlights the enlarged region in (C). **(C)** Co-localization of Brp[Nc82] and RIM[HA−Znf] in presynaptic boutons. **(D)** Whisker plots with median for Brp puncta per NMJ (normally distributed data, mean indicated by dashed line), NMJ area, and number of boutons per NMJ in rim[rescue−Znf] (n = 24 NMJs in nine animals, only 23 NMJs for boutons per NMJ) and rim[HA−Znf] (n = 22/9). Scale bars in (A) 100 μm, in (B) 10 μm, and in (C) 3 μm.

Suitable antibodies for the detection of RIM at presynaptic Drosophila AZs are still lacking. Thus, implementation of reliable RIM imaging in the fly is highly demanding, especially regarding the AZ nano-arrangement. To first probe the overall expression of RIM[HA−Znf] in the Drosophila central and peripheral nervous systems, we performed immunostainings using a monoclonal antibody against HA for detection of the N-terminal HA-tag (see the Material and Methods section) and a well-characterized, highly specific monoclonal antibody Brp[Nc82] mapping to the C-terminal region of Brp (Kittel et al, 2006; Fouquet et al, 2009; Mrestani et al, 2021; Fig 3). RIM[HA−Znf] was strongly expressed in the larval central nervous system (Fig 3A). We also detected considerable co-expression of RIM[HA−Znf] and Brp[Nc82] at third instar larval NMJs (Fig 3B and C). In addition, co-expression of RIM[V5−Znf] and Brp[Nc82] at NMJs was observed (Fig S1), although, in rim[V5−Znf] animals, baseline synaptic transmission was disturbed (Fig 2C and D). We

conclude that RIM[HA−Znf] is ubiquitously expressed in the fly central and peripheral nervous systems and specifically co-localizes with the AZ scaffold protein. Thus, super-resolution analysis of RIM[HA−Znf] at presynaptic AZs is feasible. To analyze if NMJ morphology itself is altered in rim[HA−Znf] animals, we performed immunostainings of α-HRP and Brp[Nc82] in rim[HA−Znf] and rim[rescue−Znf] (Fig 3D and Table S5). The number of Brp[Nc82] puncta per NMJ and NMJ area were unaltered, however, the number of boutons per NMJ slightly decreased in rim[HA−Znf]. We conclude that the overall NMJ morphology remains mostly unaltered in the tagged rim variant.

## Identification of RIM[HA−Znf] clusters at presynaptic AZs

Next, we performed two-color dSTORM localization microscopy of RIM[HA−Znf] at larval NMJs (Fig 4 and Table S6; Heilemann et al, 2008;

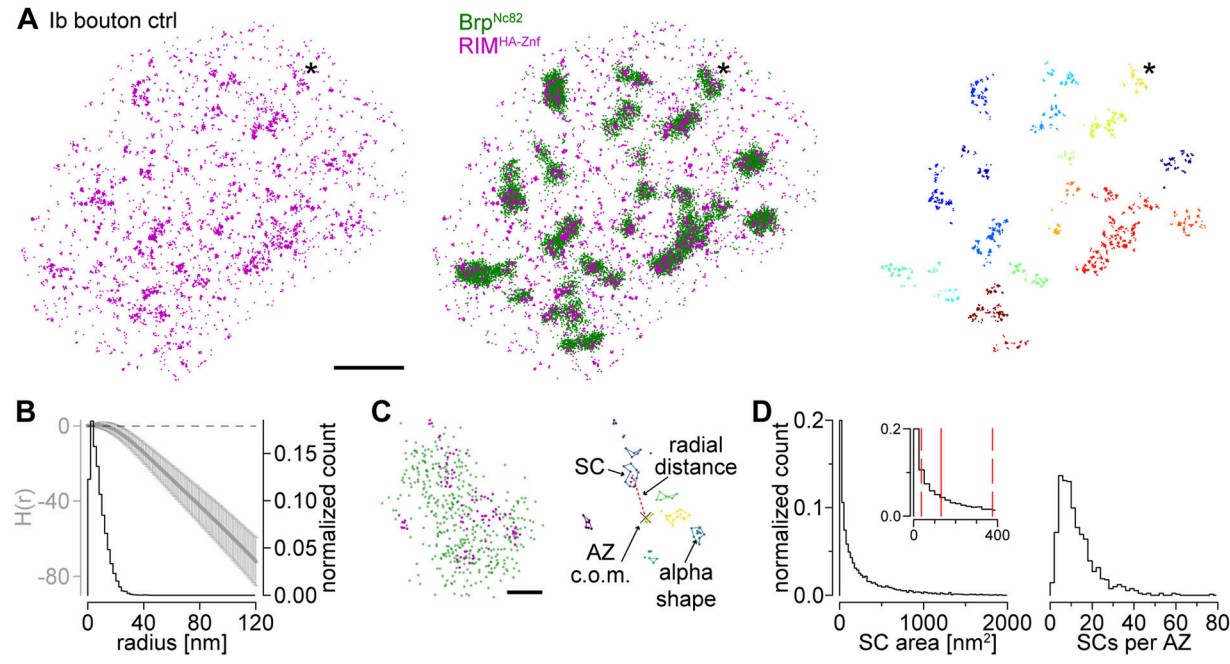

**Figure 4. RIM<sup>HA−Znf</sup> clusters at presynaptic AZs.**

**(A)** Two-channel *d*STORM localizations for a rim<sup>HA−Znf</sup> type Ib bouton. Left: RIM<sup>HA−Znf</sup> stained with α-HA antibody and Alexa Fluor647 conjugated F(ab')$_2$ fragments (magenta). Middle: overlay with Brp<sup>Nc82</sup> labeled with Alexa Fluor 532-conjugated IgGs (green). **(C)** Asterisk marks enlarged AZ in (C). Right: RIM<sup>HA−Znf</sup> localizations from left panel with all localizations with Euclidian distance >20 nm to Brp localizations removed. The removed signal is considered noise. Individual RIM<sup>HA−Znf</sup> subclusters (SCs) were extracted by HDBSCAN and assigned to nearest AZs by color. **(B)** Averaged H function (gray, mean ± SD) from n = 542 RIM<sup>HA−Znf</sup> first-level clusters obtained from 18 NMJs and nine animals (maximum of the curve indicates a mean SC radius of 6 nm) and histogram (black) of the mean radius from n = 11,094 RIM<sup>HA−Znf</sup> SCs (estimated from SC area under the assumption of a circular area, median (25$^{th}$–75$^{th}$ percentile): 6.4 (3.4–10.9) nm). Dashed black line indicates the prediction for a random Poisson distribution. **(A, C)** Enlarged plots of the AZ marked in (A). Left: two-channel *d*STORM localizations of RIM<sup>HA−Znf</sup> and Brp<sup>Nc82</sup>. Right: RIM<sup>HA−Znf</sup> SCs extracted by HDBSCAN in different colors. Colored lines indicate alpha shapes used for area determination. The center of mass (c.o.m.) of the corresponding AZ (x) is indicated. Dashed red line shows the Euclidian distance between the AZ c.o.m and an SC c.o.m., referred to as radial distance. Parameters in HDBSCAN were: minimum cluster size = 100 localizations, minimum samples = 25 localizations for Brp<sup>Nc82</sup>; minimum cluster size = 2 localizations, minimum samples = 2 localizations for RIM<sup>HA−Znf</sup>; α-value Brp<sup>Nc82</sup> = 800 nm$^2$, α-value RIM<sup>HA−Znf</sup> = 300 nm$^2$; exclusion criteria for Brp<sup>Nc82</sup> clusters were area ≤ 0.03 µm$^2$ and ≥ 0.3 µm$^2$. **(D)** Distributions of RIM<sup>HA−Znf</sup> SC area (11,094 SCs from 18 NMJs in nine animals) and the number of RIM<sup>HA−Znf</sup> SCs per AZ (n = 893 AZs from 18 NMJs in nine animals) in all AZs without selection according to AZ circularity (see the Material and Methods section). Inset in the left panel highlights the range between 0 and 400 nm$^2$ SC area. Solid red line indicates median, dashed red lines, 25$^{th}$ and 75$^{th}$ percentiles. Scale bars in (A) 1 µm, in (C) 100 nm.

van de Linde et al, 2011; Löschberger et al, 2012; Mrestani et al, 2021; Paul et al, 2022; Dannhäuser et al, 2022). Using a combination of Brp<sup>Nc82</sup> and a monoclonal antibody against the HA-tag for detection of RIM<sup>HA−Znf</sup> we observed distinct co-localization of both epitopes at presynaptic AZs of type Ib boutons (Fig 4A). Application of HDBSCAN and Ripley analyses as established in previous work (Mrestani et al, 2021; Dannhäuser et al, 2022) extracted individual RIM<sup>HA−Znf</sup> subclusters (SCs) with diameters of ~13 nm (Fig 4B and C). Using alpha shapes for area determination (Mrestani et al, 2021), we obtained RIM<sup>HA−Znf</sup> SCs of ~130 nm$^2$ size each containing about six localizations (Fig 4D and Table S6). Furthermore, an individual AZ was found to contain ~10 RIM<sup>HA−Znf</sup> SCs in about 120 nm radial distance from the AZ c.o.m. in wildtype *Drosophila* larvae (Fig 4D and Table S6). This radial distance determination included all AZs detected by the HDBSCAN algorithm independent from their orientation to the focal plane (Mrestani et al, 2021). In summary, using the HA-tag N-terminally fused to RIM, we report strong and discrete expression of RIM<sup>HA−Znf</sup> clusters in co-localization with the presynaptic scaffold protein Brp.

### Acute PHP results in compaction and addition of RIM<sup>HA−Znf</sup> subclusters

The induction of presynaptic homeostasis is associated with structural reorganization of presynaptic AZs (Weyhersmüller et al, 2011; Böhme et al, 2019; Mrestani et al, 2021). As RIM is essential for PHP expression (Müller et al, 2012), we wondered whether nanoscaled reorganization of RIM<sup>HA−Znf</sup> occurs at the *Drosophila* NMJ (Fig 5 and Tables S6 and S7). Thus, we compared Brp<sup>Nc82</sup> and RIM<sup>HA−Znf</sup> localization data using HDBSCAN-based algorithms in PhTx-treated preparations (phtx, Fig 5A) and DMSO controls (ctrl). We found no difference in RIM<sup>HA−Znf</sup> localization numbers per SC between phtx and ctrl, however, acute PHP reduced RIM<sup>HA−Znf</sup> SC areas and increased localization density (Fig 5B). Interestingly, the number of SCs per AZ along with the total number of RIM<sup>HA−Znf</sup> localizations per AZ increased in phtx (Fig 5C), corroborating earlier results from cultured murine neurons (Müller et al, 2022). The radial distance between SC c.o.m.s and the AZ c.o.m. was unchanged in phtx and the total AZ area occupied by RIM<sup>HA−Znf</sup> remained the same (Fig 5C and Table S6). We tested if the slightly decreased AZ circularity in phtx

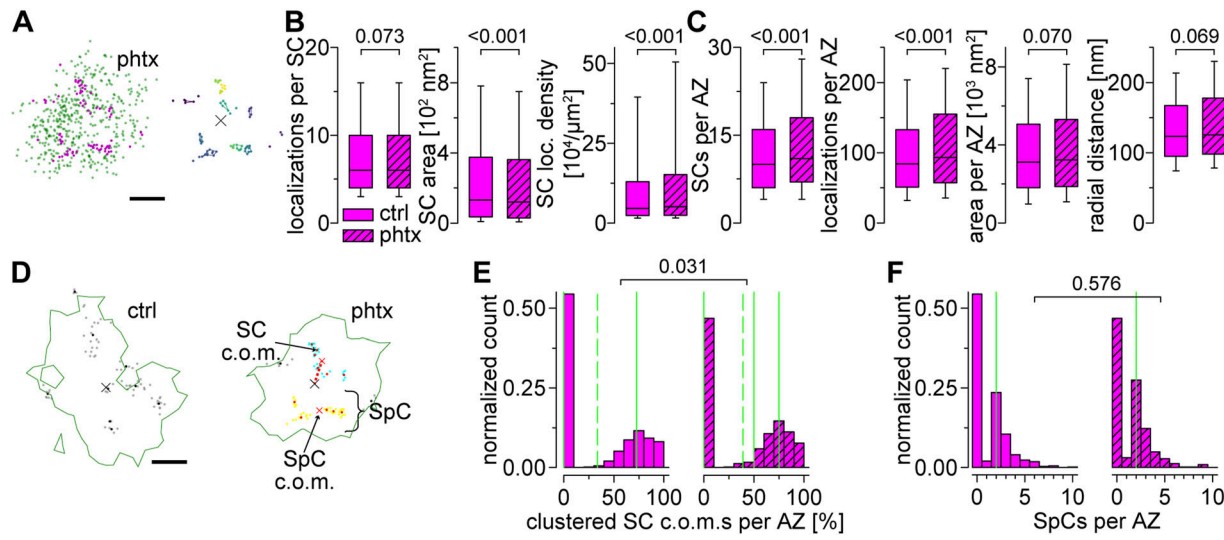

**Figure 5. RIM[HA−Znf] subclusters are recruited to SpCs in acute PHP.**
**(A)** Localizations of a type Ib AZ from a phtx larva. Left: two-channel *d*STORM data of RIM[HA−Znf] (magenta) and Brp[Nc82] (green). Right: RIM[HA−Znf] SCs extracted by HDBSCAN in different colors. Colored lines indicate alpha shapes used for area determination. The AZ c.o.m. is indicated (x). **(B)** Number of localizations per RIM[HA−Znf] SC, SC area, and SC localization density in ctrl (filled boxes, n = 11,094 SCs from 18 NMJs in nine animals) and phtx (dashed boxes, n = 13,568 SCs from 19 NMJs in 12 animals) shown as box plots (horizontal lines show median, box boundaries show 25[th] and 75[th] percentiles, whiskers show 10[th] and 90[th] percentiles). **(C)** Number of RIM[HA−Znf] SCs per AZ, the total number of RIM[HA−Znf] localizations per AZ, the entire RIM[HA−Znf] area per AZ, and radial distance of RIM[HA−Znf] SCs per AZ in ctrl (n = 893 AZs from 18 NMJs in nine animals) and phtx (n = 963 AZs from 19 NMJs in 12 animals). **(D)** Representative, circular AZs (circularity ≥ 0.6) from a ctrl and a phtx type Ib bouton. Green lines indicate alpha shapes used for AZ area determination. Left: ctrl AZ from Fig 4C with gray dots indicating individual RIM[HA−Znf] localizations and black dots indicating SC c.o.m.s that do not belong to a supercluster (SpC). Right: two SpCs (shown in blue and yellow) were extracted by HDBSCAN from the c.o.m.s (red dots) of RIM[HA−Znf] SCs in the phtx AZ. Black dots represent SC c.o.m.s that are unclustered according to HDBSCAN analysis and gray dots show localizations of the corresponding SCs. Red x indicate c.o.m.s of SpCs. Dashed red line indicates the distance between a SC c.o.m. and the AZ c.o.m. **(E)** Percentage of SC c.o.m.s that are organized in SpCs per AZ for ctrl (n = 542 AZs from 18 NMJs in nine animals) and phtx (n = 545 AZs from 19 NMJs in 12 animals) shown as histograms. Dashed lines indicate mean, solid lines indicate median, 25[th] and 75[th] percentiles. Note that the 25[th] percentile and median fall together at zero in the left panel. **(F)** Number of SpCs per AZ. Median values, indicated in green, and statistical comparison refers to AZs with at least one SpC (ctrl: n = 247 AZs from 18 NMJs in nine animals; phtx: n = 290 AZs from 19 NMJs in 12 animals). Scale bars in (A) and (D) 100 nm.

influences structural parameters (Table S6, Mrestani et al, 2021). However, all relative changes between experimental groups were present in filtered AZs in planar view indicated by AZ circularity ≥ 0.6 (Tables S6 and S7). To further control the robustness of our findings, we established an analysis routine alternative to our previous algorithm (Dannhäuser et al, 2022), now relying on HDBSCAN to account for noise. Single-channel HDBSCAN analysis of RIM[HA−Znf] localizations (Fig S2A and B) delivers less intuitive segmentation opposed to Brp[Nc82] (Fig S2C, compare Fig 1B in Mrestani et al [2021]). However, it accounts for noise in the data, as an alternate way to denoising by distance to the Brp[Nc82] signal (compare the Material and Methods section and Dannhäuser et al [2022]). Furthermore, after AZ assignment (Fig S2D), RIM[HA−Znf] SCs outside the AZ are accessible for quantification (Fig S2E and F). Whereas analysis of intrasynaptic RIM[HA−Znf] SCs confirmed compaction during PHP, no differences between ctrl and phtx were detectable for extrasynaptic SC populations (Fig S2G and Table S8). Interestingly, extrasynaptic SCs displayed similar localization numbers, increased areas, and lower localization densities opposed to their intrasynaptic counterparts (Fig S2G), implying a nanotopological differentiation of these two populations. To address whether increased RIM[HA−Znf] protein per AZ during homeostasis (Fig 5C) arises from recruitment from the AZ vicinity, we quantified the effect of PHP on RIM[HA−Znf] SC numbers and localizations in the extrasynaptic SC population in 400 nm distance around the AZ and found no difference, however, RIM[HA−Znf] SC radial distance was slightly increased (Fig S2H).

Strikingly, analyzing these parameters for intrasynaptic SCs using the two different denoising approaches delivered identical results (Figs 5C and S2I). Lastly, we employed a second level HDBSCAN analysis to investigate RIM[HA−Znf] superclusters (SpCs, Fig 5D, Dannhäuser et al, 2022). Remarkably, in this analysis, the percentage of SCs clustered into SpCs was increased after PHP (Fig 5E), and the fraction of AZs displaying superclustering at all (45.6% versus 53.32% in ctrl and phtx, respectively). Furthermore, nearest neighbor analysis revealed decreased distances between SC c.o.m.s in phtx (Fig S3A and B), suggesting enhanced clustering, that is, compaction of RIM[HA−Znf], during the homeostatic challenge. In addition, in both ctrl and phtx most AZs contained about 2–3 SpCs and about 4 SCs per SpC which displayed similar sizes (Figs 5F and S3C and D). In summary, PHP leads to more RIM[HA−Znf] SCs per AZ with increased localization density. Furthermore, the proportion of SCs clustered into SpCs is larger in phtx and distances between individual SCs are decreased, suggesting compaction of RIM[HA−Znf] during a homeostatic challenge.

## Discussion

We used CRISPR/Cas9-assisted genome engineering of *D. rim* and fused two established epitope tags N-terminally to the zinc finger domain (Fig 1). We show that both *rim* variants RIM[V5−Znf] and

RIM$^{HA-Znf}$ are efficiently expressed at presynaptic AZs in co-localization with Brp (Figs 3 and S1). Using electrophysiology and PhTx to induce PHP, we demonstrate that release is diminished in rim$^{V5-Znf}$, whereas, baseline synaptic transmission and PHP remain intact in rim$^{HA-Znf}$ (Fig 2). We determine the localization of RIM within the AZ nano-scaffold in *Drosophila*, applying a combination of two-channel localization microscopy and HDBSCAN analysis (Figs 4, S2, and S3). We detect ~10 RIM$^{HA-Znf}$ clusters per AZ of ~130 nm$^2$ size in ~120 nm distance from the AZ center, which compact and increase in numbers during acute PHP (Figs 5 and S4A and B).

### Endogenous epitope tagging of *rim*

Previous RIM imaging at AZs of *Drosophila* NMJs utilized endogenous *Minos*-mediated integration cassette-based tagging of a single splice variant (Petzoldt et al, 2020) or overexpression of a full-length N-terminally GFP-tagged fusion protein (Graf et al, 2012). While establishing a method for deciphering RIM nanoscale arrangement at the AZ, we found disturbed synaptic transmission in the GFP-tagged fusion protein (data not shown). Through the application of a previously introduced genomic editing platform (Paul et al, 2022), we fused a V5-tag inserted N-terminally to the RIM C$_2$A domain which did not yield specific imaging results (data not shown), possibly through lack of antibody accessibility of the epitope at this protein position. Therefore, we decided to focus on tag insertion at RIM's N-terminus, because this position principally worked in earlier studies (Graf et al, 2012). Furthermore, only the Rab3-binding domain is present in all *D. rim* splice variants, enabling visualization of the whole *rim* population. To reduce the possibility to disturb the RIM–Rab3 interaction via RIM's zinc finger domain through an adjacent fusion, we resorted to smaller V5 and HA epitope tags. Strikingly, whereas both peptide tags are composed of only a few amino acids, we found differences in their effect on synaptic transmission. In animals expressing the V5-tagged *rim* variant, we observed reduced evoked release, whereas RIM$^{HA-Znf}$ displayed intact neurotransmission and PHP. Reduced eEPSC amplitudes in rim$^{V5-Znf}$ animals fit with disturbed vesicle recruitment via a Rab3 interaction but the HA-tag insertion at the same position without discernible effects on synaptic function argues against these assumptions. How can the effect of the V5-tag on RIM be explained?

Both epitope tags are small and have a low molecular weight (V5: 14 amino acids, ~1.4 kD; HA: 9 amino acids, ~1.1 kD), but differ in their substituents and charged residues and, accordingly, in their isoelectric points (V5: pI = 5.84, 1 positive and 1 negative residue; HA: pI = 3.56, 2 negative residues). V5 contains more amino acids with aliphatic substituents compared with HA, which is composed of more aromatic residues. In combination with the higher isoelectric point this leads to less hydrophilicity of the V5-tag which may interfere with folding of the Rab3/Znf-domain and, thus, RIM function. In contrast, the HA-tag contains two negatively charged amino acids and, because of the aromatic residues that can form π–π interactions, the HA peptide may adopt a more compact conformation. These differences can lead to an increased hydrophilicity of the HA-tag, with less steric interactions with the protein, preserving RIM function (Fig 2). In summary, our results indicate that not only epitope tag position and size but also other, yet ill-

defined properties of the tag, profoundly influence RIM. At AZ scaffolds, RIM recruits presynaptic vesicles via Rab3 interaction to VGCCs through multiple direct and indirect interactions with its C-terminal domains (Wang et al, 1997; Kaeser et al, 2011). Assuming a central VGCC arrangement at *Drosophila* AZs (Hallermann et al, 2010; Ghelani & Sigrist, 2018; Ghelani et al, 2023), the RIM C-terminus should localize closer to AZ centers than the ~120 nm reported for N-terminal RIM SCs in the present study. Thus, simultaneous tagging and mapping of RIM N- and C-termini might be informative. However, the different *D. rim* gene products display considerable structural heterogeneity at the C-terminus with only a fraction possessing the C-terminal C$_2$B domain, inevitably leading to deviant variant subgroups during simultaneous imaging. Another tagging option is the relative mapping of RIM's N- and C-termini to VGCCs. However, the two-color *d*STORM approach as presented here using Alexa Fluor532 to establish Brp as reference signal does not meet the high-resolution requirements for such mapping studies. The described technical limitations might be overcome with more advanced dyes and more favorable photophysics and/or applying spectral demixing single-molecule localization microscopy of spectrally close far-red dyes (Lehmann et al, 2016; Wang et al, 2022).

### Rapidly reorganizing RIM nanoclusters during homeostatic plasticity

RIM is required for many forms of presynaptic plasticity, including short-term plasticity, long-term potentiation, and PHP (Castillo et al, 2002; Schoch et al, 2002; Fourcaudot et al, 2008; Pelkey et al, 2008; Müller et al, 2012). PHP is among the best characterized plasticity patterns of the *Drosophila* NMJ; however, despite emerging evidence for remarkable reorganization of RIM nanoclusters during synaptic plasticity at mammalian synapses (Tang et al, 2016), structural reorganization of RIM at *Drosophila* synapses has not been investigated yet. We report increased numbers of RIM SCs in PhTx-induced acute PHP at *Drosophila* NMJ AZs (Fig 5C), in agreement with earlier findings from cultured hippocampal neurons (Müller et al, 2022). Increased RIM amounts for enhanced vesicle traffic in PHP appear plausible regarding its role in vesicle priming and Ca$^{2+}$-triggered release (Schoch et al, 2006; Deng et al, 2011) mediated through liquid–liquid phase separation (Wu et al, 2019, 2021). However, the changes of AZ proteins in PHP appear to be differentially regulated. In earlier studies, we and others demonstrated compaction of Brp, RBP, and Unc-13 and VGCCs at AZs in PHP (Mrestani et al, 2021; Dannhäuser et al, 2022; Ghelani et al, 2023). Whereas SC areas of Brp, RBP, and RIM decreased and localization density consecutively increased, SC area and localization density of Unc-13 remained unchanged. Furthermore, Brp and Unc-13 SCs move towards the AZ center in PHP (reduced radial distance). This is a second level of compaction, and it appears that in addition to SC compaction (for some proteins), the entire AZ is compacted in PHP. Remarkably, RIM is the only epitope so far with more SCs per AZ and therefore higher overall localization numbers per AZ in PHP, consistent with a mechanism of protein recruitment to the AZ, possibly involving the kinesin-associated axonal cargo machinery (Goel et al, 2019) or altered proteostasis during homeostasis (Baccino-Calace et al, 2022). The decreased area and enhanced molecule density of RIM$^{HA-Znf}$ SCs described here (Figs 5B and S4)

fits well with the AZ compaction pattern implicated in structurally defining high release probability terminals (He et al, 2022 *Preprint*). In an earlier study, we determined a higher-level organization of Unc-13 SCs into ~2–3 SpCs per AZ matching the number of docked vesicles (Böhme et al, 2016). Protein level changes in PHP are not observed for Brp and Unc-13. Hence, increased formation of RIM[HA–Znf] SpCs during homeostasis possibly indicates recruitment of RIM[HA–Znf] SCs to nascent Unc-13-marked release sites (Fig 5D–F). This implies a dynamic AZ model in which release sites are generally defined by relatively fix Unc-13 SpCs, whereas active release sites are discernable by additionally recruited RIM SCs, supporting increased vesicle traffic.

### Perspectives on RIM nanotopology at AZ scaffolds

We provide a nanoscale quantification of the crucial AZ component RIM in *Drosophila*, utilizing a novel endogenously tagged genetic tool in combination with two-color *d*STORM and HDBSCAN algorithms. In principle, it is possible to retrieve quantitative information on protein numbers from *d*STORM data (Löschberger et al, 2012; Ehmann et al, 2014). Using Alexa Fluor 647 to quantitatively assess Brp distribution at *Drosophila* AZs, a conversion factor of 0.134 ± 0.028 (SEM) between localizations and molecules was determined (Ehmann et al, 2014). Regarding the similar staining and imaging conditions in the present study, this factor can be used for a rough approximation of RIM molecules at the AZ scaffold. Accordingly, six localizations per RIM SC may translate in ~1–2 molecules. RIM SCs are smaller than Brp, RBP, and Unc-13 SCs and at the resolution limit of our imaging approach; however, we still observe compaction in PHP. In addition, it appears promising to use this imaging strategy for studying RIM structure and function in phasic type Is versus tonic type Ib boutons. The HA-tag used in this study is incorporated in all 14 variants present in *Drosophila*; however, recent work suggested differential isoform expression in these bouton types (Jetti et al, 2023 *Preprint*). Finally, considering the remarkable structural reorganization of RIM (Yao et al, 2007; Tang et al, 2016; Müller et al, 2022), probing RIM redistribution in the context of human disease relevant point mutations, for example, the arginine to histidine substitution in the $3_{10}$-helix of the $C_2A$ domain causing human CORD7 syndrome (Paul et al, 2022), will be informative.

# Materials and Methods

## Fly stocks

*D. melanogaster* were raised on a standard cornmeal and molasses medium at 25°C. Male third instar larvae were used for experiments.

### Fly stocks generated in this study
Internal stock IDs indicated in brackets after the genotype:

*rim[ΔRab3/Znf], DsRed⁺*: *w1118*; +; *rim[ΔRab3/Znf] attP DsRed⁺/TM3, Sb*
  (RIM76, RIM98)
*rim[ΔRab3/Znf], DsRed⁻*: *w1118*; +; *rim[ΔRab3/Znf] attP DsRed⁻/TM3, Sb*
  (LAT471, LAT473)

*rim[rescue-Znf]*: *w1118*; +; *rim[ΔRab3/Znf] attP{Rab3(pLM6) mW⁻}/TM6b, Tb*
  (LAT545/AM29)
*rim[V5-Znf]*: *w1118*; +; *rim[ΔRab3/Znf] attP{V5::Rab3(pLM11) mW⁻}/TM6b, Tb*
  (LAT559/AM36)
*rim[HA-Znf]*: *w1118*; +; *rim[ΔRab3/Znf] attP{HA::Rab3(pAM62) mW⁻}/TM6b, Tb*
  (AM138; BDSC#99515).

### Other fly stocks
Source is indicated in brackets after the genotype:

*w1118* (WT, Bloomington *Drosophila* Stock Center)
*y1 M{GFP[E.3xP3]=vas-Cas9.RFP-}[ZH–2A] w1118*;;; (#55821; BDSC, Gratz et al, 2014).

### Molecular reagents

All primer sequences used in this study are listed in Table S9.

### pU6-gRNA plasmids
We used the online tool "CRISPR Optimal Target Finder" (Gratz et al, 2014) to identify Cas9 cutting sites 5′ and 3′ of the *rim* exons encoding the zinc finger domain (respective gRNAs are referred to as gRNA#1 and gRNA#2). Before cloning the presence of the cleavage sites was confirmed by Sanger sequencing of PCR products covering the respective sites (PCR primers for gRNA#1: *mh_01F/mh_02R*; for gRNA#2: *mh_03F/mh_04R*). Target-specific sequences for gRNAs (gRNA#1: GACCGACCCGGCATCTC/GTTTGG; gRNA#2: GCCTTGCGGGATACTCA/GAGCGG; sequences in 5′–3′ order, PAM site is underlined, forward slash indicates the Cas9 cutting site) were synthesized as 5′-phosphorylated oligonucleotides (gRNA#1: *mh_41F/mh_42R*; gRNA#2: *mh_43f/mh_44R*), annealed, and ligated into the *Bbs*I site of the *pU6-BbsI-chiRNA* plasmid (Gratz et al, 2013), giving rise to vectors *pMH1* (gRNA#1) and *pMH2* (gRNA#2).

### rim[ΔRab3/Znf] HDR donor plasmid
A 1-kb fragment encoding the 3′ homology arm was amplified from *w1118* genomic DNA using primers *mh_57F/mh_58R*, cut with *Sap*I, and inserted into de-phosphorylated and *Sap*I-digested *pHD-DsRed-attP* (Gratz et al, 2014), giving rise to plasmid *pMH11*. *pMH11* was *Aar*I-digested and a genomic 0.2 kb PCR product amplified with primers *mh_77F/mh_79R*, also digested with *Aar*I, was ligated generating plasmid *pMH24*. A 1.1-kb PCR product encoding the 5′ homology arm from genomic DNA was amplified with primers *mh_80F/mh_81R*, digested with *Spe*I/*Nhe*I and ligated with *Spe*I/*Nhe*I-digested *pMH24*, producing the final HDR vector *pMH10*.

### rim[rescue−Znf] attB plasmid
A 5.4-kb fragment from *w1118* genomic DNA was amplified using primers *lm_28F/lm_29R*, *Not*I/*Asc*I-digested and ligated to a *Not*I/*Asc*I-cut 6.1 kb fragment of *pGE-attB[GMR]* (Huang et al, 2009), generating the *rim[rescue-Znf]* plasmid *pLM6*.

### rim[V5−Znf] attB plasmid
A 1.1-kb fragment of genomic DNA contained by *Bst*Bl/*Xho*l sites around the *rim* start codon was synthesized by the company Thermo fisher Scientific (pTL821). This plasmid contains an additional start codon followed by the coding sequence of the 14-amino

acid V5-tag (GGTAAGCCTATCCCTAACCCTCTCCTCGGTCTCGATTCTACGC-CCGGGGGCGGCCGC; sequence in 5′–3′ order, additional *Xma*I and *Not*I sites are underlined) directly 5′ of the *rim* start codon, giving rise to the fusion of the amino acids MGKPIPNPLLGLDSTPGGGR to the RIM N-terminus. *pTL821* was *Bst*Bl/*Xho*l-digested to release the 1.1 kb fragment which was subsequently ligated into a 10.4-kb fragment of *Bst*Bl/*Xho*l-cut *pLM6*, generating the *rim^V5-Znf* vector *pLM11*.

### *rim^HA−Znf* attB plasmid

Primers *am_212F* and *am_213R* were annealed and ligated into the multiple cloning site of the *Avr*II/*Nco*I-digested subcloning vector *pMCS5*, generating plasmid *pAM60*-containing *Bst*Bl/*Xho*l sites. A 1.1-kb *Bst*Bl/*Xho*l fragment of *pLM11* was ligated into *Bst*Bl/*Xho*l-digested *pAM60*, giving rise to plasmid *pAM61*. A 0.2-kb genomic DNA fragment containing a *Hind*III site located 5′ to the *rim* start codon and the subsequent 5′-UTR followed by an additional start codon and the coding sequence of the 9 amino acid HA-tag (TACCCC-TACGACGTCCCCGACTACGCCCCCGGGGGCGGCCGC; sequence in 5′–3′ order, additional *Xma*I and *Not*I sites are underlined, results in the fusion of amino acids MYPYDVPDYAPGGGR N-terminal to the first RIM methionine) was synthesized by the company Eurofins (*pAM59*). The construct was *Hind*III/*Not*I-digested and ligated into *Hind*III/*Not*I-cut *pAM61*. The resulting subclone was digested with *Bst*Bl/*Xho*l releasing a 1.1-kb fragment that was ligated with a 10-kb *Bst*Bl/*Xho*l fragment of *pLM11*, producing the final *rim^HA-Znf* plasmid *pAM62*.

### CRISPR targeting

All transgenesis steps were performed at BestGene Inc. The gRNA plasmids (*pMH1*, *pMH2*) and the HDR donor plasmid (*pMH10*) were injected into embryos of a *Drosophila* strain with germline expression of vasa-Cas9 (Gratz et al, 2014), producing the *rim^ΔRab3/Znf*, *DsRed^+* allele. Correct transgene incorporation was confirmed by sequencing of PCR fragments covering breakpoints between genomic/transgenic DNA amplified from genomic DNA of respective adult transgenic flies and across the deleted fragment. The 3xP3-DsRed transformation marker was removed by expressing a germline Cre source and confirmed by PCR genotyping. Subsequent insertion of the different *attB* transgenes (*pLM6*, *pLM11*, *pAM62*) into *rim^ΔRab3/Znf*, *DsRed^-* by φC31-mediated transgenesis followed by Cre-driven removal of the mW+ selection marker was performed by BestGene Inc.

### Fixation, staining, and immunofluorescence

For immunofluorescence imaging of larval NMJs and ventral nerve cords (VNCs), larvae were dissected in ice-cold hemolymph-like solution (HL-3, Stewart et al, 1994), fixed with 4% PFA in PBS for 10 min, and blocked for 30 min with PBT (PBS containing 0.05% Triton X-100, Sigma-Aldrich) including 5% normal goat serum (Dianova). Primary antibodies were added for overnight staining at 4°C. After two short and three 20-min-long washing steps with PBT, preparations were incubated with secondary antibodies for 3 h at room temperature, followed by two short and three 20-min-long washing steps with PBT. Preparations were kept in PBS at 4°C until imaging.

All NMJ data were obtained from abdominal muscles 6/7 in segments A2 and A3. Directly compared data were obtained from larvae stained in the same vial and measured in one imaging session.

### Confocal microscopy and structured illumination microscopy (Apotome, SIM)

Preparation, fixation, and antibody staining were performed as described above. Primary antibodies were used in the following concentrations: mouse monoclonal α-Brp (Brp^Nc82, 1:100; AB_2314866; Developmental Studies Hybridoma Bank), rabbit monoclonal α-HA (1:500; C29F4 catalog #3724; Cell signaling technology), and mouse monoclonal α-V5 (1:100; R960-25; Invitrogen). Secondary antibodies were used in the following concentrations: goat α-rabbit conjugated Alexa Fluor 488 (1:500; A-11008; Invitrogen), goat α-mouse conjugated Alexa Fluor 488 (1:250 for SIM imaging and 1:500 for Apotome imaging; A-32723; Invitrogen), goat α-mouse conjugated Cy3 (1:500; RRID: AB_2338690; Jackson ImmunoResearch). Directly conjugated antibodies were incubated together with secondary antibodies in the following concentrations: goat α-horseradish-peroxidase (α-HRP) labeled with Alexa Fluor647 (1:500; AB_2338967; Jackson ImmunoResearch), goat α-HRP labeled with Cy3 (1:250; AB_2338959; Jackson ImmunoResearch). Larval preparations were mounted in Vectashield (Vector Laboratories). Images were acquired at room temperature. Confocal images were obtained with a Leica SP8 system (Leica Microsystems) equipped with HC PL APO 20x/0.75 IMM CORR CS2 and HC PL APO 63x/1.3 GLYC CORR CS2 objectives used for low-resolution characterization of larval VNCs and NMJs, respectively. VNCs were imaged with a z-step size of 600 nm, sum slices projections were created, and brightness and contrast were manually adjusted in FIJI. For NMJs, images were obtained with 300 nm z-step size, maximum projected, background subtracted using the rolling ball method (rolling ball radius 50 pixels) with brightness and contrast manually adjusted. To assess NMJ morphology, images were acquired using an Apotome System (Axiovert 200M, objective 63x, NA 1.4, oil; Zeiss). Brp puncta per NMJ and NMJ area were measured using a thresholding algorithm in FIJI (Schindelin et al, 2012), essentially as described previously (Mrestani et al, 2021; Paul et al, 2022). Boutons per NMJ were counted manually. SIM imaging was performed as previously described (Dannhäuser et al, 2022) using a Zeiss Elyra S.1 structured illumination microscope equipped with a sCMOS camera (pco.edge 5.5 m) and an oil-immersion objective (Plan-Apochromat 63x, 1.4 NA). Lasers with 488 and 531 nm were used. Again, images were maximum-projected and brightness and contrast were manually adjusted.

### *d*STORM

*d*STORM imaging of the specimen was performed essentially as previously reported (Ehmann et al, 2014; Paul et al, 2015, 2022; Mrestani et al, 2021; Dannhäuser et al, 2022). The same primary antibodies as described above were used in the following concentrations: mouse α-Brp (Brp^Nc82, 1:100), rabbit α-HA (1:500). The following secondary antibodies were used: goat α-rabbit F(ab')₂ fragments labeled with Alexa Fluor 647 (1:500; A21246; Thermo Fisher Scientific) and goat α-mouse IgGs labeled with Alexa Fluor 532

(1:500; A11002; Thermo Fisher Scientific). After staining, larval preparations were incubated in 100 mM mercaptoethylamine (Sigma-Aldrich) in a 0.2 M sodium phosphate buffer, pH ~7.9, to allow reversible switching of single fluorophores during data acquisition (van de Linde et al, 2008). Images were acquired on an inverted microscope (IX-71, 60x, NA 1.49, oil immersion; Olympus) equipped with a nosepiece-stage (IX2-NPS; Olympus). 647 nm (F-04306-113; MBP Communications Inc.) and 532 nm (gem 532; Laser Quantum) lasers were used for excitation of Alexa Fluor647 and Alexa Fluor532, respectively. Laser beams were passed through clean-up filters (BrightLine HC 642/10 and Semrock, ZET 532/10, respectively), combined by two dichroic mirrors (LaserMUX BS 514–543 and LaserMUX BS 473–491R, 1064R, F38-M03; AHF Analysentechnik), and directed onto the probe by an excitation dichroic mirror (HC Quadband BS R405/488/532/635, F73-832; AHF Analysentechnik). The emitted fluorescence was filtered with a quadband filter (HC-quadband 446/523/600/677; Semrock) and a long pass (Edge Basic 635; Semrock) or bandpass filter (Brightline HC 582/75; Semrock) for the red and green channels, respectively, and divided onto two cameras (iXon Ultra DU-897-U; Andor) using a dichroic mirror (HC-BS 640 imaging; Semrock). For the red channel, image resolution was 127 nm × 127 nm per pixel to obtain super-resolution of RIM$^{HA-Znf}$. For the green channel, image resolution was 130 nm × 130 nm per pixel. Localization of single fluorophores and high-resolution image reconstruction was performed with rapid*STORM* (Heilemann et al, 2008; Wolter et al, 2010; van de Linde et al, 2011; Wolter et al, 2012; https://www.biozentrum.uni-wuerzburg.de/super-resolution/archiv/rapidstorm/). Only fluorescence spots with an A/D count over 5,000 were analyzed and a subpixel binning of 10 nm px$^{-1}$ was applied.

## Analysis of localization data

Localization data were analyzed essentially as described previously (Mrestani et al, 2021; Dannhäuser et al, 2022), using custom-written Python code (https://www.python.org/, language version 3.6) and the Python interface Jupyter (Kluyver et al, 2016) to directly load and analyze localization tables from rapid*STORM*. Before the Python-based analysis, the regions of interest, corresponding to terminal six boutons, were masked in the reconstructed, binned images from rapid*STORM* using FIJI (Schindelin et al, 2012). Cluster analyses were based on the Python implementation of HDBSCAN (McInnes et al, 2017; https://github.com/scikit-learn-contrib/hdbscan) and were performed according to our previously described two-channel localization data workflow. After extracting Brp clusters in the Alexa Fluor532 channel, corresponding to individual AZs, unclustered Brp localizations were discarded from further analysis. All RIM$^{HA-Znf}$ localizations in the Alexa Fluor647 channel with an Euclidian distance > 20 nm to Brp localizations were discarded to remove noise. H functions derived from Ripley's K function were computed using Python package Astropy (Robitaille et al, 2013) according to the previously published algorithm (Dannhäuser et al, 2022) for the denoised RIM$^{HA-Znf}$ localizations and for the random Poisson distribution. Displayed curves were averaged (mean ± SD). The function was evaluated in nm steps for radii from 0 to 120 nm and without correction for edge effects. A second HDBSCAN to extract individual RIM$^{HA-Znf}$ subclusters (SCs) was performed with

parameters adjusted to yield SC radii matching the H function maximum ("minimum cluster size" = 2 localizations and "minimum samples" = 2 localizations). SCs were assigned to individual Brp clusters by selecting the lowest Euclidian distance between the center of mass (c.o.m.) of each SC and the localizations of each Brp cluster. The distance between the SC and AZ c.o.m.s is referred to as radial distance and was computed for each individual AZ as mean of the assigned RIM$^{HA-Znf}$ SCs. Cluster areas were measured using 2D alpha shapes in the Python version of CGAL (Computational Geometry Algorithms Library, https://www.cgal.org). For alpha shapes of Brp clusters and RIM$^{HA-Znf}$ SCs, we chose $\alpha$-values of 800 nm$^2$ and 300 nm$^2$, respectively. Exclusion criteria for Brp clusters were area ≤ 0.03 $\mu m^2$ and ≥ 0.3 $\mu m^2$ (Mrestani et al, 2021). RIM$^{HA-Znf}$ SC c.o.m.s that were assigned to those Brp clusters were also excluded from further analysis, and SCs where alpha shape determination failed because of a sparse signal that yielded SC areas of 0 $\mu m^2$. RIM$^{HA-Znf}$ area per AZ was computed as the sum of all RIM$^{HA-Znf}$ areas belonging to an individual AZ. Brp cluster circularity was computed as described previously (Mrestani et al, 2021). To test the robustness of the described algorithm and get access to extrasynaptic RIM$^{HA-Znf}$ localizations, we established an alternative denoising routine relying solely on HDBSCAN. Multiple combinations were tested to obtain optimal clustering parameters for single-channel HDBSCAN analysis of RIM$^{HA-Znf}$ ("minimum cluster size" = 20 localizations and "minimum samples" = 5 localizations). Unclustered RIM$^{HA-Znf}$ localizations were considered noise and discarded from further analysis. A second HDBSCAN to extract RIM$^{HA-Znf}$ SCs and Brp$^{Nc82}$ cluster assignment were performed, similarly as described above. SCs with an Euclidian distance to Brp$^{Nc82}$ localizations ≤ 50 nm (about two times median SC diameter) were considered intra-synaptic, and all other SCs were considered extrasynaptic. The extrasynaptic population with an Euclidian distance > 50 and ≤ 400 nm at AZs with circularity ≥ 0.6 is referred to as SCs in AZ vicinity. Analysis of RIM$^{HA-Znf}$ SpCs and distances between SC c.o.m.s was performed as reported in earlier work (Dannhäuser et al, 2022). Only AZs with a circularity ≥ 0.6 were selected. Python module scikit-learn (Pedregosa et al, 2011) was used to compute distances between SC c.o.m.s. To extract SpCs, HDBSCAN was performed taking the SC c.o.m.s of an individual AZ as input ("minimum cluster size" = 2 localizations and "minimum samples" = 2 localizations). The selection method was changed to "leaf" clustering. For statistical comparison of SpC numbers per AZ between experimental groups, only AZs where SpCs could be detected were included. The SpC c.o.m. was defined as the c.o.m. of its respective SC c.o.m.s and the Euclidean distance between these SC c.o.m.s and the SpC c.o.m. was computed as mean per SpC. Data used for 2D models of idealized type Ib AZs are summarized in Table S10. Circle radii for Brp$^{Nc82}$ area were calculated from previously reported data (Mrestani et al, 2021, ctrl: r = 186.3 nm, phtx: r = 179.3 nm). SC sizes are displayed to scale. SCs were randomly positioned by hand between 10$^{th}$ and 90$^{th}$ percentiles of the radial distance of the respective epitopes.

## Electrophysiology

Two-electrode voltage clamp recordings (Axoclamp 2B amplifier, Digidata 1440A; Molecular Devices) were obtained from abdominal muscle 6 in segments A2 and A3 as previously described

(Dannhäuser et al, 2022; Paul et al, 2022). All measurements were obtained at room temperature in HL-3 (Stewart et al, 1994) with the following composition (in mM): NaCl 70, KCl 5, $MgCl_2$ 20, $NHCO_3$ 10, trehalose 5, sucrose 115, Hepes 5, and $CaCl_2$ 1, pH adjusted to 7.2. Intracellular electrodes had resistances of 10–30 MΩ and were filled with 3 M KCl. For analysis, only cells with an initial membrane potential of –50 mV or less and a membrane resistance of ≥ 4 MΩ were included. During recordings, cells were clamped at a holding potential of –80 mV (miniature EPSCs, mEPSCs) or –60 mV (evoked EPSCs, eEPSCs). Signals were low-pass filtered at 10 kHz and analyzed in Clampfit (Version 11.1, Molecular Devices). mEPSCs were recorded for 90 s and the occurrence rate of mEPSCs determined mEPSC frequency. Amplitude, rise time, and decay time constants were determined using an average of all mEPSCs recorded within one time period. To evoke synaptic currents, nerves were stimulated via a suction electrode with pulses of 300 $\mu$s length and typically at 12 V (Grass S48 stimulator and isolation unit SIU5; Astro-Med). We applied a paired-pulse protocol with 0.2 Hz frequency and 30 ms interpulse intervals. For analysis, 5–10 traces per interval were averaged. The quantal content was estimated by dividing the mean eEPSC amplitude by the mean mEPSC amplitude measured in one cell. mEPSC amplitudes were corrected for the more hyperpolarized holding potential (Hallermann et al, 2010).

### PhTx treatment

PhTx 433 tris (trifluoroacetate) salt (PhTx, CAS 276684-27-6; Santa Cruz Biotechnology) was dissolved in dimethyl sulfoxide (DMSO) to obtain a stock solution of 4 mM and stored at –20°C. For each experiment, the respective volume was further diluted with freshly prepared HL-3 to a final PhTx concentration of 20 $\mu$M in 0.5% DMSO. Control experiments were performed with the same DMSO concentration in HL-3. PhTx treatment of semi-intact preparations was performed essentially as described previously (Frank et al, 2006; Mrestani et al, 2021; Dannhäuser et al, 2022). In brief, larvae were pinned down in calcium-free, ice-cold HL-3 at the anterior and posterior endings, followed by a dorsal incision along the longitudinal axis. Larvae were incubated in 10 $\mu$l of 20 $\mu$M PhTx in DMSO for 10 min at room temperature. After this incubation time, PhTx was replaced by HL-3 and larval preparations were completed, followed by electrophysiological measurements or *d*STORM imaging.

### Statistics

Statistical analyses were performed with Sigma Plot 13 (Systat Software) or GraphPad Prism 9. D'Agostino & Pearson (electrophysiology) or Shapiro–Wilk (imaging data) were used to test normality. If data were not normally distributed, we used the nonparametric Mann–Whitney rank sum test (eEPSC and mEPSC amplitudes in rim[HA–Znf] ctrl versus phtx; *d*STORM parameters of RIM[HA–Znf] and Brp[Nc82] in rim[HA–Znf] ctrl versus phtx), the Kruskal–Wallis test (mEPSC frequency, mEPSC rise time, eEPSC amplitude, eEPSC rise time, paired pulse ratios of wt, rim[rescue–Znf], rim[V5–Znf] and rim[HA–Znf]) or one-way ANOVA on ranks (multiple comparisons of extra- and intrasynaptic SC populations, followed by pairwise comparisons using Dunn's method) and reported data as median

(25th–75th percentile), if not indicated otherwise. Normally distributed data were analyzed using a two-tailed *t* test (quantal content in rim[HA–Znf] ctrl versus phtx, Brp puncta per NMJ in rim[rescue–Znf] versus rim[HA–Znf]) or one-way ANOVA (mEPSC amplitude, mEPSC tau decay, eEPSC tau decay in wt, rim[rescue–Znf], rim[V5–Znf], and rim[HA–Znf]) and reported as mean ± SEM. In box plots, horizontal lines represent median, boxes, quartiles, and whiskers 10th and 90th percentiles, unless indicated otherwise. Scatter plots show individual data points. Bin counts in histograms were normalized to the total number of observed events which was set to 1. All plots were produced with Sigma Plot. Figures were assembled using Adobe Illustrator (Adobe Creative Cloud 2022). Tables S1–S8 and S10 contain all numerical values stated or not stated in text and figure legends including *P*-values and sample sizes.

## Data Availability

The authors declare that the custom-written Python code and all datasets supporting the findings of this work are available from the corresponding authors.

## Supplementary Information

## Acknowledgements

This work was supported by grants from the German Research Foundation (DFG) FOR 3004 SYNABS P1 to M Heckmann, the IZKF Würzburg to M Heckmann (N229) and MM Paul (Z-3/69 and HB22PAUL), the CRC 1423 project number 421152132/projects A06 and B06 to T Langenhan, and the University of Leipzig Clinician Scientist Program and Jung Foundation for Science and Research through Jung Career Advancement Prize 2023 to A Mrestani. The authors thank M Oppmann and F Köhler for technical assistance.

### Author Contributions

A Mrestani: conceptualization, data curation, formal analysis, funding acquisition, investigation, visualization, and writing—original draft.
S Dannhäuser: data curation, formal analysis, investigation, and writing—review and editing.
M Pauli: software, formal analysis, validation, and investigation.
P Kollmannsberger: software, formal analysis, and writing—review and editing.
M Hübsch: investigation.
L Morris: investigation.
T Langenhan: conceptualization, funding acquisition, and writing—review and editing.
M Heckmann: conceptualization, resources, supervision, funding acquisition, validation, methodology, and writing—original draft and project administration.

MM Paul: conceptualization, data curation, formal analysis, supervision, funding acquisition, validation, investigation, visualization, and writing—original draft and project administration.

## Conflict of Interest Statement

The authors declare that they have no conflict of interest.

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
