## [Reviewer comments · Life Science Alliance]

Life Science Alliance

Nanoscaled RIM clustering at presynaptic active zones revealed by endogenous tagging

Achmed Mrestani, Sven Dannhäuser, Martin Pauli, Philip Kollmannsberger, Martha Hübsch, Lydia Morris, Tobias Langenhan, Manfred Heckmann, and Mila Paul

DOI: <https://doi.org/10.26508/lsa.202302021>

Corresponding author(s): Mila Paul, University of Würzburg and Manfred Heckmann, University of Würzburg

Review Timeline:

Submission Date:	2023-03-02
Editorial Decision:	2023-03-30
Revision Received:	2023-06-27
Editorial Decision:	2023-07-18
Revision Received:	2023-08-27
Accepted:	2023-08-28

Scientific Editor: Novella Guidi

Transaction Report:

March 30, 2023

Re: Life Science Alliance manuscript #LSA-2023-02021-T

Dr. Mila Marie Paul
University of Würzburg
Röntgenring 9
Würzburg 97070
Germany

Dear Dr. Paul,

Thank you for submitting your manuscript entitled "Nanoscaled RIM clusters at presynaptic release sites of *Drosophila melanogaster*" to Life Science Alliance. The manuscript was assessed by expert reviewers, whose comments are appended to this letter. We invite you to submit a revised manuscript addressing the Reviewer comments.

Thank you for this interesting contribution to Life Science Alliance. We are looking forward to receiving your revised manuscript.

Sincerely,

B. MANUSCRIPT ORGANIZATION AND FORMATTING:

Reviewer #1 (Comments to the Authors (Required)):

Mrestani and Dannhäuser et al. report the development of a novel HA-tagged version of the AZ protein RIM in *Drosophila*. The RIM-HA localizes to AZs and does not interfere with synaptic function. By contrast, a V5-tagged RIM version disrupts baseline synaptic function. The authors then use the HA-tagged RIM protein for dSTORM microscopy followed by hierarchical clustering. Results indicate clustered localization of RIM at *Drosophila* AZs. Upon expression of presynaptic homeostatic plasticity (PHP), RIM cluster numbers increase, but have smaller area, indicating a 'compaction'-like reconfiguration. An additional clustering step suggests that upon PHP, RIM clusters are more likely to arrange in so-called superclusters.

Developing tools to study the localization of AZ proteins without overexpression is of high importance for our understanding of synaptic structure and function. The presented HA-tagged RIM will thus prove useful for future studies. The genetic and functional experiments are well performed and convincing. The clustering analysis of RIM is interesting but may benefit from additional controls. Investigating the dynamic re-arrangement of AZ components during synaptic plasticity is of high relevance. However, the present dSTORM results reported upon PhTx treatment are less convincing and the conclusions drawn are not fully supported by the data.

I have the following specific comments for the authors:

- 1) The present study is largely reminiscent of a previous work by the authors on the AZ protein UNC13 (Dannhäuser et al. 2022). It would be interesting to discuss how the results for the two (and other?) proteins compare during PHP.
- 2) Did the authors check if the tagged RIM constructs affect NMJ morphology? A comparison to wt or the rescue condition could be added.
- 3) For the analysis in Figure 4, any signal >20nm away from Brp was removed (and considered noise). An advantage of HDBSCAN is that it can account for noise in the data. What would the result of HDBSCAN be without excluding data?
- 4) The results on structural changes during PHP do not appear to be very strong. The authors observe quite small changes, such as on average one more RIM SC per AZ upon PhTx or a decrease of 10 nm² for their area. The data raise the question if results are still observable when individual NMJs or animals are considered as experimental unit (N) instead of AZs. Furthermore, what is the biological significance of such small changes (<10%)? It may also be helpful to indicate effect sizes.
- 5) The final analysis of RIM SC "superclusters" is also not very strong. First, the fraction of AZs showing a supercluster configuration seems unaltered after PhTx, and so is the number of superclusters per AZ and the number of SCs per supercluster. The only observed difference lies in the percentage of SCs that are organized in superclusters per AZ. The authors conclude that "[PhTx] enhanced clustering", which is not directly evident from these data. Yet, this finding is based on statistical analysis of zero-inflated data (Figure 5E). Indeed, the second level HDBSCAN analysis revealed superclusters in less than half of AZs. This should be mentioned. Could the authors use statistical analysis methods that are more suited for take zero-inflated data? Otherwise, I suggest analyzing the AZs with superclusters separately and clearly stating the fraction of AZs without superclusters (is the fraction different?). And interpreting the results regarding an effect on clustering should take these considerations into account.
- 6) Figure 5 could benefit from a slightly different presentation of the data. I would find it helpful to show representative AZ images for both conditions in panel a. Further, the color-code could be improved (condition color is similar to RIM color, and the color fill of text labels is not well visible) - maybe gray for ctrl and color for PhTx? In panel E, the bin size seems rather large given the number of samples. For panel F, why does the label read "ctrl"? And I find it unusual to illustrate the median as a box - wouldn't a line be more appropriate? Finally, are boxplots indeed the best way to represent the data (given the large number of samples)?
- 7) The caption of Figure 5 is not well supported by the data. Without a release site marker, it is difficult to conclude that RIM SCs are recruited to release sites.
- 8) Have the authors studied PHP in the V5-tagged RIM variant? If the V5 tag interferes with the Zn finger domain function, this

could be insightful.

9) Some minor points in the Methods section as follows:

- Line 193: "normal goat serum" is probably more common.
- It would be helpful to state the parameters used for HDBSCAN.
- Line 284/302: ""minimum samples" two": it is unclear what "two" refers to.
- Line 314: should probably read NaHCO₃.
- Line 316: "membrane potential of at least -50 mV" is slightly misleading (assuming the authors refer to cells being more negative than this cutoff).
- Line 318: "minis": for consistency within the manuscript, I suggest using "mEPSCs" instead.
- Please specify how mEPSCs were detected and analyzed.
- Line 324: "cells were given 10 s rest", but the previous sentence refers to a stimulation frequency of 0.2 Hz. Please clarify.
- Line 339f: "larvae were incubated in 10 ul of 20 uM PhTx": The incubation volume seems very small (10 ul).
- Statistics: It is unclear which data were analyzed using parametric tests. It appears to be a minority; in which case the authors could consider using non-parametric testing throughout.

Reviewer #2 (Comments to the Authors (Required)):

The authors Mrestani et al. set out in this manuscript to endogenously tag all isoforms of the active zone scaffold RIM using established CRISPR/Cas9 approaches in *Drosophila*. As no antibodies exist for immunostaining against *Drosophila* RIM, and there is evidence that previous GFP-tagged transgenes/alleles work sub-optimally, there is a need for such a resource for the field to really understand where this crucial active zone component localizes relative to other proteins and importantly how it is remodeled during plasticity. The authors demonstrate that their HA-tagged allele localizes as expected and does not perturb synaptic function or PHP plasticity. As in previous studies, the authors go on to use super resolution dSTORM imaging to show that during PHP, HA-RIM becomes "compacted" at AZs and analyze spatial properties of RIM relative to BRP.

Overall this is an important resource for the field as the HA-RIM appears to tag all RIM isoforms, important when considered with recent RNAseq work showing differential isoforms at release sites (see below). There are also some intriguing findings related to RIM's change in plasticity, which in contrast to what the authors have shown for BRP and Unc13, RIM may not only become compacted but also increase in abundance. There are few areas detailed below that the authors should consider to strengthen the conclusions and expand the insights from this study, but this is still a well done study that will be of interest to a variety of researchers.

Main comments:

1. RIM PHP changes vs BRP and Unc13: Details were unclear to this reviewer regarding how RIM changes at AZs after PhTx application relative to what these authors have already shown happen for BRP and Unc13. While it seems all 3 AZ proteins become "compacted" at individual AZs, including an increase in density, there seem to be some interesting differences with RIM. In particular, the authors made relatively strong conclusions in previous studies that while density of BRP and Unc13 increased, there was no net change in overall abundance. However, if I understand the results of RIM in this study, not only does PHP induce RIM compaction, but there is also an apparent increase in the abundance of RIM. Can the author clarify more fully and explicitly in the discussion how RIM may (or may not) behave differently during PHP relative to previous studies and discuss implications?
2. Non-BRP RIM signals at AZs: In Fig. 4A and elsewhere, the authors show dSTORM localizations of both BRP and RIM at individual Ib boutons. While BRP localizations are very well restricted to individual AZ sites, RIM signals appear enriched at BRP areas, but signals are also clear at extra-AZ areas of the boutons. Is this signal just background from the HA-staining, or do the authors think these may be extra-AZ reservoirs of RIM? It would be very interesting to quantify this non-BRP RIM signal and determine if it changes after PhTx application, which may provide a pool of RIM to be mobilized to AZs during PHP.
3. Of interest but may be outside the scope: There are a number of areas that additional experiments would provide insights of interest to the field but are not absolutely essential. First, understanding RIM localization at AZs at the convergent Is input may be interesting to contrast with RIM structure at the larger Ib AZs, including relative to BRP; the authors and others have already characterized other AZ components at both Ib and Is AZs. This would be particularly interesting since differential RIM isoforms have been suggested to be expressed in Is vs Ib AZs from a recent RNAseq study from Troy Littleton's lab.

In addition, observing whether the same changes in RIM are observed in GluRIIA mutants as they have demonstrated in PhTx, would also be of interest. However, since both of these questions have been studied with other components in previous studies I don't think they are essential for this study.

Minor comments:

1. It would be useful to show all expected splice isoforms of RIM or confirm the tagged location is encoded in each of the 14 putative isoforms. Even more informative would be to define whether there are distinct tagged RIM splice variants expressed in Is vs Ib motor neurons as suggested by a recent study by Troy Littleton using RNAseq. This might also be informative to discuss

in the manuscript.

2. The title does not really capture the key findings of the study. Maybe something like "Endogenous tagging of RIM reveals nanoscaled clustering at release sites".
3. The title of the subsection at lines 382-83 do not accurately reflect their findings - transmission is not unaltered in rim-V5.
4. NMJ structure/growth: It would be useful to count boutons or assess NMJ morphology in the new rim alleles generated in this study to ensure no significant changes are observed.
5. Lines 407-409 should be rewritten for clarity.

Reviewer #3 (Comments to the Authors (Required)):

The manuscript by Mrestani et al describes the generation of new transgenic lines to visualize the active zone protein RIM in *Drosophila*. The authors use a combination of CRISPR/Cas9 and phiC31 integrase to place V5 and HA epitopes to the N-terminus of RIM. Using electrophysiological analysis, the authors find the HA tag is well tolerated, but the V5 tag disrupts synaptic transmission. Using the HA-tagged RIM protein, the authors then use STORM imaging to examine the normal distribution of RIM and how its localization changes during acute induction of presynaptic homeostatic plasticity (PHP) by addition of Phtx toxin. The authors find RIM co-localizes with the active zone protein BRP as expected, and can undergo compaction during PHP, consistent with the reorganization that has been described for BRP. The authors also find some increase in RIM subcluster number that suggest more RIM protein may also be accumulating at the active zone during PHP. Overall, the work provides a new tool to analyze RIM localization and suggest it acts similarly to the active zone protein BRP in undergoing compaction during this form of homeostatic synaptic plasticity.

Major comments:

1. Where do the authors think the excess RIM is coming from that accumulates at the active zone during PHP. Is there a decrease in RIM levels outside of active zones that matches the increase?
2. The effects to RIM localization/clustering shown in Figure 5B, C, E and F appear very mild, with quite subtle changes in the mean values between control and Phtx-treated larvae. How do these changes compare to what the authors previously observed with BRP during PHP? Are these of similar magnitude, or instead a smaller effect?
3. The authors estimate 10-20 RIM molecules per active zone. What would be the actual change during PHP - an addition of 1 or 2 RIM proteins, or something more dramatic? It can be hard to go from the STORM imaging changes to what the model is for RIM redistribution and accumulation. Indeed, a supplementary model figure would be helpful in this regards.

Minor comments:

1. Lines 469-473, the structure of the sentence is confusing.
2. In Fig 4D(ii) the color of the dashed line to indicate the radial distance is the same as the color c.o.m of the RIM SCs, which makes it difficult to distinguish the two.

Reviewer #1 (Comments to the Authors (Required)):

Mrestani and Dannhäuser et al. report the development of a novel HA-tagged version of the AZ protein RIM in *Drosophila*. The RIM-HA localizes to AZs and does not interfere with synaptic function. By contrast, a V5-tagged RIM version disrupts baseline synaptic function. The authors then use the HA-tagged RIM protein for dSTORM microscopy followed by hierarchical clustering. Results indicate clustered localization of RIM at *Drosophila* AZs. Upon expression of presynaptic homeostatic plasticity (PHP), RIM cluster numbers increase, but have smaller area, indicating a 'compaction'-like reconfiguration. An additional clustering step suggests that upon PHP, RIM clusters are more likely to arrange in so-called superclusters.

Developing tools to study the localization of AZ proteins without overexpression is of high importance for our understanding of synaptic structure and function. The presented HA-tagged RIM will thus prove useful for future studies. The genetic and functional experiments are well performed and convincing. The clustering analysis of RIM is interesting but may benefit from additional controls. Investigating the dynamic re-arrangement of AZ components during synaptic plasticity is of high relevance. However, the present dSTORM results reported upon PhTx treatment are less convincing and the conclusions drawn are not fully supported by the data.

We thank the reviewer for the careful assessment of our manuscript and the constructive and helpful criticism for improvement. Following the recommendations, we added a new experiment to our manuscript (Figure 3D, Table S6) and carried out further data analyses resulting in two new Supplementary Figures 2 and 4 plus related Supplementary Tables S9 and 10. Furthermore, we critically revised several text passages, redesigned details within our Figures and corrected some wording in Tables S4-S8. We address the major points raised by the reviewer as follows in detail.

I have the following specific comments for the authors:

1) The present study is largely reminiscent of a previous work by the authors on the AZ protein UNC13 (Dannhäuser et al. 2022). It would be interesting to discuss how the results for the two (and other?) proteins compare during PHP.

Following the suggestion, we specifically compared the AZ proteins studied during Philanthotoxin induced PHP using super-resolution in this and previous studies (Mrestani et al., 2021; Dannhäuser et al., 2022). We summarize the results comparing Brp, RBP, Unc-13 and RIM in the following table:

	SC area	loc. No per SC	loc. Density	SC numbers	radial distance	overall area	overall locs
Brp	↓	-	↑	-	↓	↓	-
RBP	↓	-	↑	ND	ND	ND	ND
Unc-13	-	-	-	-	↓	↓	↓
RIM	↓	-	↑	↑	-	-	↑

↓ = decreased, ↑ = increased, - = unchanged, ND = not determined

To further discuss these aspects, we added a new section to our Discussion (ll. 615-623): “Whereas SC areas of Brp, RBP and RIM decreased and localization density consecutively increased, SC area and localization density of Unc-13 remained unchanged. Furthermore, Brp, Unc-13 and RIM SCs move towards the AZ center in PHP (reduced radial distance). This is a second level of compaction and it appears that in addition to SC compaction (for some proteins) the entire AZ appears to be compacted in PHP. Remarkably, RIM is the only epitope so far with more SCs per AZ and therefore higher overall localization numbers per AZ in PHP, consistent with a mechanism of protein recruitment to the AZ or altered proteostasis during homeostasis (Baccino-Calace et al., 2022).”

2) Did the authors check if the tagged RIM constructs affect NMJ morphology? A comparison to wt or the rescue condition could be added.

We performed new experiments to investigate whether the RIM constructs affect NMJ morphology. We compared NMJs of $\text{rim}^{\text{rescue-Znf}}$ and $\text{rim}^{\text{HA-Znf}}$ larvae. NMJs were imaged employing α -HRP against presynaptic membranes and AZs were detected using Brp^{Nc82} (Kittel et al., 2006; Paul et al., 2022). We compared NMJ areas, the number of boutons per NMJ and the number of Brp puncta per NMJ (equaling AZ numbers) in both genotypes. We included these data adding a new panel D to Figure 3 (II. 460-465, II. 710-713).

3) For the analysis in Figure 4, any signal > 20nm away from Brp was removed (and considered noise). An advantage of HDBSCAN is that it can account for noise in the data. What would the result of HDBSCAN be without excluding data?

To address the reviewer's comment, we carried out further analyses using HDBSCAN algorithms. We added a new Figure S2 and a text paragraph describing these results (II. 502-522): "To further control the robustness of our findings we established an analysis routine alternative to our previous algorithm (Dannhäuser et al., 2022), now relying on HDBSCAN to account for noise. Single-channel HDBSCAN analysis of RIMHA-Znf localizations (Fig. S2 A, B) delivers less intuitive segmentation opposed to BrpNc82 (Fig. S2 C, compare Fig. 1 B in Mrestani et al., 2021). However, it accounts for noise in the data, as an alternate way to denoising by distance to the BrpNc82 signal (compare Material and Methods and Dannhäuser et al., 2022). Furthermore, after AZ assignment (Fig. S2 D) RIMHA-Znf SCs outside the AZ are accessible for quantification (Fig. S2 E, F). While analysis of intrasynaptic RIMHA-Znf SCs confirmed compaction during PHP, no differences between ctrl and phtx were detectable for extrasynaptic SC populations (Fig. S2 G, Table S9). Interestingly, extrasynaptic SCs displayed similar localization numbers, increased areas and lower localization densities opposed to their intrasynaptic counterparts (Fig. S2 G), implying a nanotopological differentiation of these two populations. To address whether increased RIMHA-Znf protein per AZ during homeostasis (Fig. 5 C) arises from recruitment from the AZ vicinity, we quantified the effect of PHP on RIMHA-Znf SC numbers and localizations in the extrasynaptic SC population in 400 nm distance around the AZ and found no difference, however, RIMHA-Znf SC radial distance was slightly increased (Fig. S2 H). Strikingly, analyzing these parameters for intrasynaptic SCs using the two different denoising approaches delivered identical results (Fig. 5 C and Fig. S2 I)."

4) The results on structural changes during PHP do not appear to be very strong. The authors observe quite small changes, such as on average one more RIM SC per AZ upon PhTx or a decrease of 10 nm² for their area. The data raise the question if results are still observable when individual NMJs or animals are considered as experimental unit (N) instead of AZs. Furthermore, what is the biological significance of such small changes (<10%)? It may also be helpful to indicate effect sizes.

We agree with the reviewer that the structural changes during PHP appear not to be very strong. The uncertainty which is the correct statistical reference for 'n' is a challenge. Of the three possibilities 1) n = AZ, 2) n = image or 3) n = animal each comprises a degree of data pooling (Mrestani et al., 2021). In previous work we computed a linear mixed model treating PHP as a fixed effect and the differences between NMJs as a random effect. This model confirmed the differences found for statistics computed with 'n = AZ' (compare Figure S7 and Supplementary Table S1D in Mrestani et al., 2021). We repeated a pooled analysis for our data using 'n = recorded image' for statistical tests and summarize the results in the table below. AZ populations per image could be pooled by computing means or medians. As data in 'n = image' were not all normally distributed, we use 'image means' and 'image medians'. In addition, we performed this analysis using the entire dataset (i.e., 'all AZs' within the table) and using only AZs with high circularity (i.e., 'AZs with circularity ≥ 0.6 '); compare Material and

Methods for details). This analysis revealed a loss of statistical significance for some parameters, while others remained significantly different. However, like in a previous analysis of Brp data (Mrestani et al., 2021), statistical power implies that this approach is underpowered (power < 0.8) and likely underestimates the effects. Pooling the results of individual NMJs before applying statistical tests discards a substantial amount of information in the data. Additionally, images contain different numbers of AZs, and pooling their AZs as means or medians likely leads to unreliable statistical estimates. In conclusion, we believe that using 'n = AZs' for our statistics describes the properties and the underlying distribution of the data better than more extensive pooling. Regarding biological plausibility of the numerically small structural changes reported here, we would like to refer to an earlier data simulation approach which demonstrated that moderate changes in 2D localization data translate into larger changes in 3D molecule concentration (compare Figure 5D in Mrestani et al., 2021).

parameter	ctrl	phtx	p-value	power
RIM^{HA-Znf} ctrl (n = 18 images) vs. phtx (n = 19 images)				
all AZs				
image means				
locs. per SC	8 (8-9)	8 (8-8)	0.354	
SC area [nm ²]	308 ± 36	289 ± 28	0.089 ^A (0.045) ^B	0.397 (0.527)
SC loc. density [locs./μm ²]	217,128 (180,258-314,421)	283,656 (222,130-455,836)	0.032	
SCs per AZ	11 (9-16)	13 (11-17)	0.197	
locs. per AZ	92 (83-135)	100 (86-145)	0.281	
area per AZ [nm ²]	3,434 (3,125-4,686)	3,553 (3,154-4,665)	0.659	
radial distance [nm]	137 (129-141)	144 (128-157)	0.075	
image medians				
locs. per SC	6 (6-6)	6 (6-6)	0.089	
SC area [nm ²]	142 ± 34	127 ± 27	0.162	0.285
SC loc. density [locs./μm ²]	43,984 (38,459-50,306)	46,467 (41,108-53,968)	0.294	
SCs per AZ	10 (8-13)	12 (8-15)	0.259	
locs. per AZ	84 (68-113)	87 (77-125)	0.386	
area per AZ [nm ²]	3,243 ± 732	3,419 ± 1039	0.556	0.089
radial distance [nm]	122 (117-130)	128 (119-144)	0.242	
AZs with circularity ≥ 0.6				
image means				
locs. per SC	8 (8-9)	8 (8-9)	0.354	
SC area [nm ²]	309 ± 34	286 ± 26	0.023	0.639
SC loc. density [locs./μm ²]	211,886 (138,913-333,504)	278,782 (187,773-594,641)	0.050	
SCs per AZ	11 ± 4	11 ± 3	0.385	0.137
locs. per AZ	81 (68-111)	91 (74-121)	0.574	
area per AZ [nm ²]	3,148 (2,766-3,928)	3,181 (2,706-3,931)	0.915	
radial distance [nm]	119 ± 11	118 ± 14	0.946	0.051
image medians				
locs. per SC	6 (6-6)	6 (6-6)	0.028	
SC area [nm ²]	150 ± 37	129 ± 30	0.066 ^A (0.033) ^B	0.456 (0.586)

SC loc. density [locs./ μm^2]	41,101 (35,988-49,330)	47,007 (41,380-55,101)	0.062	
SCs per AZ	9 \pm 3	10 \pm 3	0.370	0.143
locs. per AZ	68 (60-95)	77 (65-104)	0.605	
area per AZ [nm^2]	2,526 (2,234-3,239)	2,828 (1,988-3,307)	0.704	
radial distance [nm]	111 \pm 9	114 \pm 13	0.366	0.145

For statistics for n = image using all AZs or only AZs with circularity ≥ 0.6 data per image were pooled by computing either image means or medians, and data are presented for both versions. For normally distributed data (i.e., Shapiro-Wilk and Brown-Forsythe test passed) a one-tailed t-test (marked by superscription ^A) and a two-tailed t-test (marked by superscription ^B) were performed and data are reported as mean \pm SD. For not normally distributed data a Mann-Whitney Rank Sum test was used and results reported as median (25th-75th percentile). Statistical power for t-tests implies that this approach is underpowered (i.e., with power < 0.8) and thus, likely underestimates the effects because pooling the results of individual NMJs before applying statistical tests discards information.

5) The final analysis of RIM SC "superclusters" is also not very strong. First, the fraction of AZs showing a supercluster configuration seems unaltered after PhTx, and so is the number of superclusters per AZ and the number of SCs per supercluster. The only observed difference lies in the percentage of SCs that are organized in superclusters per AZ. The authors conclude that "[PhTx] enhanced clustering", which is not directly evident from these data. Yet, this finding is based on statistical analysis of zero-inflated data (Figure 5E). Indeed, the second level HDBSCAN analysis revealed superclusters in less than half of AZs. This should be mentioned. Could the authors use statistical analysis methods that are more suited for take zero-inflated data? Otherwise, I suggest analyzing the AZs with superclusters separately and clearly stating the fraction of AZs without superclusters (is the fraction different?). And interpreting the results regarding an effect on clustering should take these considerations into account.

We thank the reviewer for alerting us to this aspect. The presentation of the results from supercluster (SpC) analysis (Figure 5) was not optimal. Thus, we reworked our SpC analysis. We calculated the fraction of AZs with superclustering: 45.6 % in ctrl vs. 53.32 % in phtx. SpCs are a property in more than half of the AZs after PHP induction (but not under baseline condition, see revised manuscript II. 524-526). We would like to point out that this was not evident in originally presented histograms in Figure 5E due to erroneous scaling of the (not displayed) y axis in the right panel. We corrected this error and now the first bin height in the revised Figure 5E shows the results correctly. Next, we compared the percentage of SCs organized into SpCs omitting the zero data, i.e., only for AZs that contained SpCs, and found no difference between the experimental groups (75 (63-87) % in ctrl vs. 73 (64-85) % in phtx, respectively, p = 0.626 for n = 247 and 290 AZs, respectively). However, since the fractions of AZs with SpCs are clearly different between the groups, we believe that this analysis discards an important part of the underlying distribution. We disagree with the reviewer that zero data are necessarily a meaningless part of the distribution. Moreover, we believe that particularly in this case, the 'zero data' carry biological meaning, i.e., that PHP induction increases the probability of SpC formation.

6) Figure 5 could benefit from a slightly different presentation of the data. I would find it helpful to show representative AZ images for both conditions in panel a. Further, the color-code could be improved (condition color is similar to RIM color, and the color fill of text labels is not well visible) - maybe gray for ctrl and color for PhTx? In panel E, the bin size seems rather large given the number of samples. For panel F, why does the label read "ctrl"? And I find it unusual to illustrate the median as a box - wouldn't a line be more appropriate? Finally, are boxplots indeed the best way to represent the data (given the large number of samples)?

We revised the design of Figure 5. Regarding panel A however, a representative example AZ for ctrl is already shown in Figure 4C (corresponding to the ctrl example in Fig. 5D). Thus, we consider another example shown here in Figure 5A of minor importance. We also changed the color code. As we used magenta for all $\text{rim}^{\text{HA-Znf}}$ plots in this manuscript we favor its continuation here for both ctrl and phtx (as both are the same genotype, i.e., $\text{rim}^{\text{HA-Znf}}$). In addition, we consider grey suboptimal for the ctrl group as it is used for $\text{rim}^{\text{rescue-Znf}}$ in Fig. 1 and 2. We suggest to use magenta with a black frame for both ctrl and phtx and additionally crosshatch the phtx boxes and bars in panels B, C, E and F. For further simplification, the label 'phtx' in panel A and D was changed to black letters. In panel E, we changed bin sizes to 5% instead of the initially used 10% and provide the new panels for verification by the reviewer here (see Figure below). Nevertheless, we kindly disagree with the reviewer that a smaller bin size is advantageous and suggest to keep the 10% bins. The label 'ctrl SpCs per AZ' in panel F is wrong and should be 'SpCs per AZ'. We corrected this in the revised Figure. Regarding panel F, the reviewer suggests using a line for the median instead of a box. We changed panel F accordingly, as well as panel E for consistency. Finally, we are aware that box plots have some disadvantages regarding data illustration, especially as single values are somehow neglected or presented insufficiently. However, we strongly favor the application of box plots in particular because the large amount of data needs a certain reduction to provide adequate transport of information.

Histograms displaying the percentage of SC c.o.m.s organized into SpCs per AZ with 5 % bins.

7) The caption of Figure 5 is not well supported by the data. Without a release site marker, it is difficult to conclude that RIM SCs are recruited to release sites.

We changed the Figure caption to “RIM^{HA-Znf} subclusters are recruited to RIM SpCs in acute PHP”.

8) Have the authors studied PHP in the V5-tagged RIM variant? If the V5 tag interferes with the Zn finger domain function, this could be insightful.

This is interesting. We haven't studied PHP in the V5-tagged RIM variant. The main reason is that we are not entirely sure regarding the mechanism of disturbance, i.e., how exactly does the N-terminal V5-tag interfere with the usual interactions of the Zn-finger domain for example with the synaptic vesicle. As our knowledge regarding this molecular mechanism is limited so far, we have hesitated to perform an experiment creating data with unsure interpretability. Nevertheless, deciphering the exact mechanism leading to the molecular inference of Zn-finger function is an interesting experiment for the future, however, we believe beyond the scope of this manuscript.

9) Some minor points in the Methods section as follows:

- Line 193: "normal goat serum" is probably more common.

“natural goat serum” was changed into “normal goat serum”

- It would be helpful to state the parameters used for HDBSCAN.

To address this aspect, we summarized all HDBSCAN parameters used in this study adding them to the legend of Figure 4 (ll. 734-738). These are namely:

minimum cluster size = 2 localizations

minimum samples = 2 localizations

α -value Brp^{Nc82} = 800 nm² and α -value RIM^{HA-Znf} = 300 nm²

Exclusion criteria for Brp^{Nc82} clusters were area $\leq 0.03 \mu\text{m}^2$ and $\geq 0.3 \mu\text{m}^2$. In addition, the parameters are explained in the Methods section > Analysis of localization data (II. 286 ff.).

- Line 284/302: ""minimum samples" two": it is unclear what "two" refers to. **We agree with the reviewer that the phrasing was imprecise. The sentence refers to the assignment of the numerical value to each parameter, i.e., the value for minimum cluster size was 2 and the value for minimum samples as well. Thus, we changed the wording in the manuscript to: "minimum cluster size" = 2 localizations and "minimum samples" = 2 localizations. (I. 286 and II. 303-304)**

- Line 314: should probably read NaHCO₃. **NaHCO³ was corrected to NaHCO₃**

- Line 316: "membrane potential of at least -50 mV" is slightly misleading (assuming the authors refer to cells being more negative than this cutoff). **We agree. The wording "at least -50 mV" was changed into "-50 mV or less".**

- Line 318: "minis": for consistency within the manuscript, I suggest using "mEPSCs" instead. **We abolished the term "minis" and now write "miniature EPSCs, mEPSCs" instead.**

- Please specify how mEPSCs were detected and analyzed. **We added the description to the Material and Methods section > Electrophysiology stating the following: "Signals were low-pass filtered at 10 kHz and analyzed in Clampfit (Version 11.1, Molecular Devices). mEPSCs were recorded for 90 seconds and the occurrence rate of mEPSCs determined mEPSC frequency. Amplitude, rise time and decay time constants were determined using an average of all mEPSCs recorded within one time period." (II. 337-341)**

- Line 324: "cells were given 10 s rest", but the previous sentence refers to a stimulation frequency of 0.2 Hz. Please clarify. **The correct "resting time" was 5 s. To clarify this issue in the revised version of our manuscript we deleted the two sentences "Paired-pulse recordings were performed with interstimulus intervals (ISI) of 30 ms. To assess basal synaptic transmission 10 EPSCs evoked at 0.2 Hz were averaged per cell. Between recordings, cells were given a 10 s rest." replacing them by the new sentence "We applied a paired-pulse protocol with 0.2 Hz frequency and 30 ms interpulse intervals."**

- Line 339f: "larvae were incubated in 10 ul of 20 uM PhTx": The incubation volume seems very small (10 ul). **The incubation volume (10 μl) is indeed small; however, the authors approved this volume in multiple test experiments using different incubation volumes and found a small volume working the best in our hands.**

- Statistics: It is unclear which data were analyzed using parametric tests. It appears to be a minority; in which case the authors could consider using non-parametric testing throughout. **We used different statistical tests depending on the data distribution (parametric or non-parametric), which in our opinion is the most detailed data refinement. We strongly favor maintaining this, nevertheless, to further clarify this issue, we revised the text passage in our Material and Methods > Statistics section now stating the following: "If data were not normally distributed, we used the non-parametric Mann-Whitney Rank Sum test (eEPSC and mEPSC amplitudes in rim^{HA-Znf} ctrl vs. phtx; dSTORM parameters of RIM^{HA-Znf} and Brp^{Nc82} in rim^{HA-Znf} ctrl vs. phtx), the Kruskal-Wallis test (mEPSC frequency, mEPSC rise time, eEPSC amplitude, eEPSC rise time, PPR of wt, rim^{rescue-Znf}, rim^{V5-Znf} and rim^{HA-Znf}) or one-way ANOVA on Ranks (multiple comparisons of extra- and intrasynaptic SC populations, followed by pairwise comparisons using Dunn's Method) and reported**

data as median (25th-75th percentile), if not indicated otherwise. Normally distributed data were analyzed using a two-tailed t-test (quantal content in rim^{HA-Znf} ctrl vs. phtx, Brp puncta per NMJ in rim^{rescue-Znf} vs. rim^{HA-Znf}) or one-way ANOVA (mEPSC amplitude, mEPSC tau decay, eEPSC tau decay in wt, rim^{rescue-Znf}, rim^{V5-Znf} and rim^{HA-Znf}) and reported as mean ± SEM.” (II. 366-377)

Reviewer #2 (Comments to the Authors (Required)):

The authors Mrestani et al. set out in this manuscript to endogenously tag all isoforms of the active zone scaffold RIM using established CRISPR/Cas9 approaches in Drosophila. As no antibodies exist for immunostaining against Drosophila RIM, and there is evidence that previous GFP-tagged transgenes/alleles work sub-optimally, there is a need for such a resource for the field to really understand where this crucial active zone component localizes relative to other proteins and importantly how it is remodeled during plasticity. The authors demonstrate that their HA-tagged allele localizes as expected and does not perturb synaptic function or PHP plasticity. As in previous studies, the authors go on to use super resolution dSTORM imaging to show that during PHP, HA-RIM becomes "compacted" at AZs and analyze spatial properties of RIM relative to BRP.

Overall, this is an important resource for the field as the HA-RIM appears to tag all RIM isoforms, important when considered with recent RNAseq work showing differential isoforms at release sites (see below). There are also some intriguing findings related to RIM's change in plasticity, which in contrast to what the authors have shown for BRP and Unc13, RIM may not only become compacted but also increase in abundance. There are few areas detailed below that the authors should consider to strengthen the conclusions and expand the insights from this study, but this is still a well-done study that will be of interest to a variety of researchers.

We would like to thank the reviewer for assessment of our manuscript and the valuable suggestions for improving its quality. We added a new experiment to our manuscript (Figure 3D, Table S6) and carried out further data analyses resulting in two new Supplementary Figures 2 and 4 plus related Supplementary Tables S9 and 10. Furthermore, we critically revised several text passages, redesigned details within our Figures and corrected some wording in Tables S4-S8. Following the reviewer's recommendations, we revised our manuscript for improving its quality and address the concerns as explained in detail below.

Main comments:

1. RIM PHP changes vs BRP and Unc13: Details were unclear to this reviewer regarding how RIM changes at AZs after PhTx application relative to what these authors have already shown happen for BRP and Unc13. While it seems all 3 AZ proteins become "compacted" at individual AZs, including an increase in density, there seem to be some interesting differences with RIM. In particular, the authors made relatively strong conclusions in previous studies that while density of BRP and Unc13 increased, there was no net change in overall abundance. However, if I understand the results of RIM in this study, not only does PHP induce RIM compaction, but there is also an apparent increase in the abundance of RIM. Can the author clarify more fully and explicitly in the discussion how RIM may (or may not) behave differently during PHP relative to previous studies and discuss implications?

As we studied different AZ proteins during PHP using super-resolution to data, namely Brp, RBP, Unc-13 and RIM using either antibody pairs or endogenous tagging, we compared the different results for these molecules summarizing the important findings:

	SC area	loc. no per SC	loc. density	SC numbers	radial distance	overall area	overall locs
Brp	↓	-	↑	-	↓	↓	-

RBP	↓	-	↑	ND	ND	ND	ND
Unc-13	-	-	-	-	↓	↓	↓
RIM	↓	-	↑	↑	-	-	↑

↓ = decreased, ↑ = increased, - = unchanged, ND = not determined

We added a new section to our Discussion (ll. 615-623): “Whereas SC areas of Brp, RBP and RIM decreased and localization density consecutively increased, SC area and localization density of Unc-13 remained unchanged. Furthermore, Brp, Unc-13 and RIM SCs move towards the AZ center in PHP (reduced radial distance). This is a second level of compaction and it appears that in addition to SC compaction (for some proteins) the entire AZ appears to be compacted in PHP. Remarkably, RIM is the only epitope so far with more SCs per AZ and therefore higher overall localization numbers per AZ in PHP, consistent with a mechanism of protein recruitment to the AZ or altered proteostasis during homeostasis (Baccino-Calace et al., 2022).”

2. Non-BRP RIM signals at AZs: In Fig. 4A and elsewhere, the authors show dSTORM localizations of both BRP and RIM at individual Ib boutons. While BRP localizations are very well restricted to individual AZ sites, RIM signals appear enriched at BRP areas, but signals are also clear at extra-AZ areas of the boutons. Is this signal just background from the HA-staining, or do the authors think these may be extra-AZ reservoirs of RIM? It would be very interesting to quantify this non-BRP RIM signal and determine if it changes after PhTx application, which may provide a pool of RIM to be mobilized to AZs during PHP.

Following the reviewer’s suggestion, we analyzed our data regarding the RIM^{HA-Znf} staining outside of AZs. We summarized these findings in a new Figure S2. Furthermore, we kindly refer to the answer to reviewer #1 point 3.

3. Of interest but may be outside the scope: There are a number of areas that additional experiments would provide insights of interest to the field but are not absolutely essential. First, understanding RIM localization at AZs at the convergent Is input may be interesting to contrast with RIM structure at the larger Ib AZs, including relative to BRP; the authors and others have already characterized other AZ components at both Ib and Is AZs. This would be particularly interesting since differential RIM isoforms have been suggested to be expressed in Is vs Ib AZs from a recent RNAseq study from Troy Littleton's lab.

We agree that understanding RIM localization at phasic type Is AZs will be of interest, especially in comparison with the here presented RIM localization data at tonic type Ib AZs. The HA-tag used for RIM imaging in this study was expressed in all RIM isoforms (see below, minor comment #1). Nevertheless, the differential expression reported in Jetti et al., 2023 is interesting with regard to distinct effects in both bouton types at the Drosophila NMJ. Thus, we added this thought into our Discussion section as we agree with the reviewer that this is an interesting experiment, but outside the scope of this present study. We now state in the revised version of our manuscript: “In addition, it appears promising to use this imaging strategy for studying RIM structure and function in phasic type Is vs. tonic type Ib boutons. The HA-tag used in this study is incorporated in all 14 variants present in *Drosophila*, however, recent work suggested differential isoform expression in these bouton types (Jetti et al., 2023).” (ll. 647-651)

In addition, observing whether the same changes in RIM are observed in GluRIIA mutants as they have demonstrated in PhTx, would also be of interest. However, since both of these questions have been studied with other components in previous studies, I don't think they are essential for this study.

We thank the reviewer for raising this point. The suggested experiment analyzing RIM in GluRIIA-ko of *Drosophila* serving as an established model for chronic homeostasis is interesting. Nevertheless, repeating the experiments presented for acute homeostasis in this manuscript using the GluRIIA-ko would require time-consuming crossing experiments which we consider beyond the scope of this work. In addition,

structural changes during acute and chronic homeostasis seem to resemble (Weyhersmüller et al., 2011; Mrestani et al., 2021).

Minor comments:

1. It would be useful to show all expected splice isoforms of RIM or confirm the tagged location is encoded in each of the 14 putative isoforms. Even more informative would be to define whether there are distinct tagged RIM splice variants expressed in Is vs Ib motor neurons as suggested by a recent study by Troy Littleton using RNAseq. This might also be informative to discuss in the manuscript.

The HA-tag was expressed in all 14 RIM isoforms as shown in the Figure below. We added a paragraph within the Discussion section in the revised version of our manuscript mentioning this information, especially with regard to the differential expression of RIM between type Is and type Ib boutons. (ll. 647-651).

2. The title does not really capture the key findings of the study. Maybe something like "Endogenous tagging of RIM reveals nanoscaled clustering at release sites".

We changed the title to "Nanoscaled RIM clustering at presynaptic active zones of *Drosophila melanogaster* revealed by endogenous tagging".

3. The title of the subsection at lines 382-83 do not accurately reflect their findings - transmission is not unaltered in rim-V5.

We changed the title of the subsection accordingly to: "Baseline synaptic transmission at rim^{V5-Znf} and rim^{HA-Znf} terminals" (l. 407)

4. NMJ structure/growth: It would be useful to count boutons or assess NMJ morphology in the new rim alleles generated in this study to ensure no significant changes are observed.

We thank the reviewer for raising this aspect. As we also consider the question of NMJ structure important for our study, we carried out further experiments and included the data in the new version of our manuscript (Figure 3D, Table S6). We also kindly refer to the answer no. 2 to reviewer #1 regarding further details.

5. Lines 407-409 should be rewritten for clarity.

We changed the wording in ll. 431-433 to the following: "Thus, we probed if our genetically engineered rim variants carrying an epitope tag at the N-terminal Zn finger domain still exhibit PHP at normal levels. To test if the HA-tagged RIM is still functional we (...)"

Reviewer #3 (Comments to the Authors (Required)):

The manuscript by Mrestani et al describes the generation of new transgenic lines to visualize the active zone protein RIM in *Drosophila*. The authors use a combination of CRISPR/Cas9 and phiC31 integrase to place V5 and HA epitopes to the N-terminus of RIM. Using electrophysiological analysis, the authors find the HA tag is well tolerated, but the V5 tag

disrupts synaptic transmission. Using the HA-tagged RIM protein, the authors then use STORM imaging to examine the normal distribution of RIM and how its localization changes during acute induction of presynaptic homeostatic plasticity (PHP) by addition of Phtx toxin. The authors find RIM co-localizes with the active zone protein BRP as expected, and can undergo compaction during PHP, consistent with the reorganization that has been described for BRP. The authors also find some increase in RIM subcluster number that suggest more RIM protein may also be accumulating at the active zone during PHP. Overall, the work provides a new tool to analyze RIM localization and suggest it acts similarly to the active zone protein BRP in undergoing compaction during this form of homeostatic synaptic plasticity.

We thank the reviewer for the careful assessment of our manuscript and valuable suggestions for improvement. We added a new experiment to our manuscript (Figure 3D, Table S6) and carried out further data analyses resulting in two new Supplementary Figures 2 and 4 plus related Supplementary Tables S9 and 10. Furthermore, we critically revised several text passages, redesigned details within our Figures and corrected some wording in Tables S4-S8. In summary, we address the major points raised by the reviewer as follows in detail.

Major comments:

1. Where do the authors think the excess RIM is coming from that accumulates at the active zone during PHP. Is there a decrease in RIM levels outside of active zones that matches the increase?

One of the outstanding effects of RIM reorganization in PHP are the increased numbers of RIM localizations at presynaptic AZs (compare answers to point 1 from reviewer #1 and reviewer #2, respectively). The question where this additional RIM originates from is crucial. We carried out further data analyses without data de-noising, as explained in detail in the answers to point 3 from reviewer #1 and point 2 from reviewer #2. These data show no evidence for RIM recruitment from the close vicinity of the AZ. This might be due to technical reasons, as in this data-set the focal plane of our dSTORM imaging was chosen according to the Brp^{Nc82} signal clearly localizing in several tens of nm distance from the presynaptic membrane, as the Brp^{Nc82} is localized in this part of the Brp protein itself. Nevertheless, the question where the additional RIM signal in PHP comes from remains unsolved and surely should be addressed in further work.

2. The effects to RIM localization/clustering shown in Figure 5B, C, E and F appear very mild, with quite subtle changes in the mean values between control and Phtx-treated larvae. How do these changes compare to what the authors previously observed with BRP during PHP? Are these of similar magnitude, or instead a smaller effect?

We thank the reviewer for this important remark. The changes between rim^{HA-Znf} ctrl and phtx larvae are mild but significant. When comparing the amount of change between Brp and rim^{HA-Znf} in proportional changes, the difference between the two groups is larger in rim^{HA-Znf} animals: whereas Brp SC areas decreased to 96% (1,677 to 1617 nm²), rim^{HA-Znf} SC areas decreased to 92% (130 to 120 nm²) and whereas the Brp localization density in SCs increased to 102% (33,914 to 34,290 locs/μm²) the rim^{HA-Znf} SC localization density increased to 109% (46,582 to 50,951 locs/μm²). Thus, the effects presented in the current study are larger. Furthermore, we suggest that the fundamental underlying challenge is the uncertainty which is the correct statistical reference for 'n'. Of the three possibilities i) n = AZ, ii) n = recording, iii) n = animal each only comprises some degree of pooling the data (Mrestani et al., 2021). For further details we refer to the answer provided to reviewer #1 point 4.

3. The authors estimate 10-20 RIM molecules per active zone. What would be the actual change during PHP - an addition of 1 or 2 RIM proteins, or something more dramatic? It can be hard to go from the STORM imaging changes to what the model is for RIM redistribution and accumulation. Indeed, a supplementary model figure would be helpful in these regards.

To illustrate the molecular changes observed by dSTORM imaging, we created model type Ib AZs of ctrl and phtx. These show modifications of Brp and RIM taking previous

data and the data from this manuscript into consideration (Mrestani et al., 2021). We included a new Supplementary Figure 4 to the manuscript (II. 1147-1154):

Figure S4. AZ changes in PHP deciphered by super-resolution imaging. (A, B) Model of a type Ib AZ before (A, ctrl) and after induction of PHP (B, phtx) based on data acquired with *d*STORM and HDBSCAN analysis. Brp^{Nc82} (green) and RIM^{HA-Znf} (magenta) were imaged as described in Mrestani et al., 2021 and the present study and imaging data served for numerical values in the model. Note compaction of the entire AZ area and RIM^{HA-Znf} SCs as well as RIM^{HA-Znf} SpC formation during PHP. Brp SCs and their compaction are not shown for clarity. Scale bar 100 nm.

Minor comments:

1. Lines 469-473, the structure of the sentence is confusing.

We changed the sentence to the following: “The radial distance between SC c.o.m.s and the AZ c.o.m. was unchanged in phtx and the total AZ area occupied by RIM^{HA-Znf} remained the same (Figure 5C, Table S7).” (II. 497-499)

2. In Fig 4D(ii) the color of the dashed line to indicate the radial distance is the same as the color c.o.m of the RIM SCs, which makes it difficult to distinguish the two.

We changed the color of the dashed line in Figure 4Dii from black to red.

Literature:

- Baccino-Calace, M., Schmidt, K., and Müller, M. 2022. The E3 ligase Thin controls homeostatic plasticity through neurotransmitter release repression. *eLife* 11:e71437. <https://doi.org/10.7554/eLife.71437>
- Dannhäuser, S., A. Mrestani, F. Gundelach, M. Pauli, F. Komma, P. Kollmannsberger, M. Sauer, M. Heckmann, and M.M. Paul. 2022. Endogenous tagging of Unc-13 reveals nanoscale reorganization at active zones during presynaptic homeostatic potentiation. *Front. Cell. Neurosci.* 16:1074304.
- Jetti, S.K., Crane, A.B., Akbergenova, Y., Aponte-Santiago, N.A., Cunningham, K.L., Whittaker, C.A., and Littleton, J.T. 2023. Molecular Logic of Synaptic Diversity Between *Drosophila* Tonic and Phasic Motoneurons. *bioRxiv* January 19, 2023. <https://doi.org/10.1101/2023.01.17.524447>
- Kittel, R.J., C. Wichmann, T.M. Rasse, W. Fouquet, M. Schmidt, A. Schmid, D.A. Wagh, C. Pawlu, R.R. Kellner, K.I. Willig, S.W. Hell, E. Buchner, M. Heckmann, S.J. and Sigrist. 2006. Bruchpilot promotes active zone assembly, Ca²⁺ channel clustering, and vesicle release. *Science* 312:1051-1054.
- Mrestani, A., M. Pauli, P. Kollmannsberger, F. Repp, R.J. Kittel, J. Eilers, S. Doose, M. Sauer, A.L. Sirén, M. Heckmann, and M.M. Paul. 2021. Active zone compaction correlates with presynaptic homeostatic potentiation. *Cell Rep.* 37:109770.
- Paul, M.M., S. Dannhäuser, L. Morris, A. Mrestani, M. Hübsch, J. Gehring, G.N. Hatzopoulos, M. Pauli, G.M. Auger, G. Bornschein, N. Scholz, D. Ljaschenko, M. Müller, M. Sauer, H. Schmidt, R.J. Kittel, A. DiAntonio, I. Vakonakis, M. Heckmann, and T. Langenhan. 2022. The human cognition-enhancing *CORD7* mutation increases active zone number and synaptic release. *Brain* 12:awac011.
- Weyhersmüller, A., S. Hallermann, N. Wagner, and J. Eilers. 2011. Rapid active zone remodeling during synaptic plasticity. *J. Neurosci.* 31:6041-6052.

July 18, 2023

RE: Life Science Alliance Manuscript #LSA-2023-02021-TR

Dr. Mila Marie Paul
University of Würzburg
Röntgenring 9
Würzburg 97070
Germany

Dear Dr. Paul,

Thank you for submitting your revised manuscript entitled "Nanoscaled RIM clustering at presynaptic active zones revealed by endogenous tagging". We would be happy to publish your paper in Life Science Alliance pending final revisions necessary to meet our formatting guidelines.

- please address the final Reviewer 2's points
- please upload your main manuscript text as an editable doc file
- please upload all figure files as individual ones, including the supplementary figure files; all figure legends should only appear in the main manuscript file
- please remove figures from the manuscript text
- please add a Summary Blurb/Alternate Abstract to our system
- please add the Twitter handle of your host institute/organization as well as your own or/and one of the authors in our system
- please note that the titles in the system and on the manuscript file must match
- please consult our manuscript preparation guidelines <https://www.life-science-alliance.org/manuscript-prep> and make sure your manuscript sections are in the correct order
- please add your main, supplementary figure, and table legends to the main manuscript text after the references section;
- please use the [10 author names et al.] format in your references (i.e., limit the author names to the first 10)
- please upload your Tables in editable .doc or excel format;
- there is a callout for Figure 1D, although the figure doesn't have a D panel - please correct
- please add callouts for Figures S3D, S4A-B to your main manuscript text

A. FINAL FILES:

-- Summary blurb (enter in submission system): A short text summarizing in a single sentence the study (max. 200 characters including spaces). This text is used in conjunction with the titles of papers, hence should be informative and complementary to the title. It should describe the context and significance of the findings for a general readership; it should be written in the

present tense and refer to the work in the third person. Author names should not be mentioned.

B. MANUSCRIPT ORGANIZATION AND FORMATTING:

Sincerely,

Reviewer #1 (Comments to the Authors (Required)):

I would like to thank the authors for their clarifications as well as their additional experiments and analyses. My previous points have been adequately addressed.

Reviewer #2 (Comments to the Authors (Required)):

The authors have done a good job of responding to my central questions and concerns and those of the other reviewers in this revised manuscript. I think this study is an important contribution to the field, and the endogenously tagged Rim allele will be a powerful reagent for future studies. I do, however, have a few points for the authors to consider regarding proper discussion and scholarship of their findings in context with other important studies in the field:

1. The point about Rim increasing in abundance is an interesting one, and appears to be unique in the AZ machinery studied so far by this group, where Brp, Rbp, and Unc13 only seem to "compact" but not increase in abundance. The question of how PHP remodels AZ machinery, and whether there is an enhancement in abundance of these components, is a question that has been a topic of considerable discussion in the field. The authors should be sure to fairly cite the relevant papers and contributions to this debate (i.e. PMID: 3069227; 36800417; 30842428; 30914419, others).

2. Further, regarding my question and that of R3 about where the additional Rim is coming from during PHP, this question has been the subject of important previous studies regarding other AZ components (Brp, etc). Importantly, a couple of axonal motors (e.g., Arl-8, etc) have been suggested to mobilize pools of AZ (and SV) components, and disruption of these motors blocks PHP expression. It appears these studies were essentially ignored by the authors in this revised manuscript. Given that the authors now show evidence for enhanced Rim after PHP, I strongly suggest the authors cite and discuss these relevant studies in the final revised manuscript (see PMID: 30914419; 30842428).

Reviewer #3 (Comments to the Authors (Required)):

The authors have modified the text and performed additional analysis in the revised manuscript. Although I'm not expert in the STORM analysis methods that were brought up by Reviewer 1, I'm satisfied with the authors' revisions. Although the effect size of RIM changes in PHP are small, the toolkits they created will be useful for the field, along with the general insights into RIM localization.

Reviewer #1 (Comments to the Authors (Required)):

I would like to thank the authors for their clarifications as well as their additional experiments and analyses. My previous points have been adequately addressed.

We thank the reviewer for the repeated review and positive evaluation of our manuscript.

Reviewer #2 (Comments to the Authors (Required)):

The authors have done a good job of responding to my central questions and concerns and those of the other reviewers in this revised manuscript. I think this study is an important contribution to the field, and the endogenously tagged Rim allele will be a powerful reagent for future studies. I do, however, have a few points for the authors to consider regarding proper discussion and scholarship of their findings in context with other important studies in the field:

We thank the reviewer for the evaluation of our revised manuscript and appreciate the opinion after renewed examination. We address the open aspects in detail as follows below.

1. The point about Rim increasing in abundance is an interesting one, and appears to be unique in the AZ machinery studied so far by this group, where Brp, Rbp, and Unc13 only seem to "compact" but not increase in abundance. The question of how PHP remodels AZ machinery, and whether there is an enhancement in abundance of these components, is a question that has been a topic of considerable discussion in the field. The authors should be sure to fairly cite the relevant papers and contributions to this debate (i.e., PMID: 3069227; 36800417 30842428; 30914419).

The reviewer is right to state the studies mentioned above merit to be discussed alongside the question how PHP remodels AZs. Thus, we added two citations to our manuscript: "Assuming a central VGCC arrangement at *Drosophila* AZs (Hallermann et al., 2010; Ghelani and Sigrist, 2018; Ghelani et al., 2023) the RIM C-terminus should localize closer to AZ centers than the ~120 nm reported for N-terminal RIM SCs in the present study." (ll. 300-303) and "However, the changes of AZ proteins in PHP appear to be differentially regulated. In earlier studies we and others demonstrated compaction of Brp, RBP and Unc-13 as well as VGCCs at AZs in PHP (Mrestani et al., 2021; Dannhäuser et al., 2022; Ghelani et al., 2023)." (ll. 327-330) and "Remarkably, RIM is the only epitope so far with more SCs per AZ and therefore higher overall localization numbers per AZ in PHP, consistent with a mechanism of protein recruitment to the AZ, possibly involving the kinesin-associated axonal cargo machinery (Goel et al., 2019), or altered proteostasis during homeostasis (Baccino-Calace et al., 2022)." (ll. 336-340).

2. Further, regarding my question and that of R3 about where the additional Rim is coming from during PHP, this question has been the subject of important previous studies regarding other AZ components (Brp, etc). Importantly, a couple of axonal motors (e.g., Arl-8, etc) have been suggested to mobilize pools of AZ (and SV) components, and disruption of these motors blocks PHP expression. It appears these studies were essentially ignored by the authors in this revised manuscript. Given that the authors now show evidence for enhanced Rim after PHP, I strongly suggest the authors cite and discuss these relevant studies in the final revised manuscript (see PMID: 30914419; 30842428).

We absolutely agree with the reviewer that the question where the additional RIM is coming from during PHP remains to be resolved and discussed. However, we would

like to point to the studies mentioned by the reviewer, which are Böhme et al., 2019 and Goel et al., 2019 and essentially referenced within our manuscript (i.e., ll. 203, 344 and 339).

Reviewer #3 (Comments to the Authors (Required)):

The authors have modified the text and performed additional analysis in the revised manuscript. Although I'm not expert in the STORM analysis methods that were brought up by Reviewer 1, I'm satisfied with the authors' revisions. Although the effect size of RIM changes in PHP are small, the toolkits they created will be useful for the field, along with the general insights into RIM localization.

We are thankful for the reviewer's opinion regarding our work and value the repeated thorough and positive evaluation.

August 28, 2023

RE: Life Science Alliance Manuscript #LSA-2023-02021-TRR

Dr. Mila Marie Paul
University of Würzburg
Röntgenring 9
Würzburg 97070
Germany

Dear Dr. Paul,

Thank you for submitting your Research Article entitled "Nanoscaled RIM clustering at presynaptic active zones revealed by endogenous tagging". It is a pleasure to let you know that your manuscript is now accepted for publication in Life Science Alliance. Congratulations on this interesting work.

DISTRIBUTION OF MATERIALS:

Again, congratulations on a very nice paper. I hope you found the review process to be constructive and are pleased with how the manuscript was handled editorially. We look forward to future exciting submissions from your lab.

Sincerely,
